# SOLVING ROBUST MDPs THROUGH NO-REGRET DYNAMICS

## ABSTRACT

Reinforcement Learning is a powerful framework for training agents to navigate different situations, but it is susceptible to changes in environmental dynamics. However, solving Markov Decision Processes that are robust to changes is difficult due to nonconvexity and complex interactions between policy and environment. While most works have analyzed this problem by taking different assumptions on the problem, a general and efficient theoretical analysis is still missing. We generate a simple, Nonconvex No-Regret framework for improving robustness by solving a minimax iterative optimization problem where a policy player and an environmental dynamics player are playing against each other. By decoupling the behavior of both players with our framework, we yield several scalable algorithms that solve Robust MDPs under different conditions on the order of $\mathcal{O}\left(\frac{1}{T^{\frac{1}{2}}}\right)$ with only a convex uncertainty set assumption.

Reinforcement Learning (RL) is a powerful subset of machine learning that enables agents to learn through trial-and-error interactions with their environment. RL has succeeded in various applications such as game playing, robotics, and finance (Sutton & Barto, 2018). However, when a trained policy operates in different environmental dynamics than the training environment, it often underperforms and achieves suboptimal rewards (Farebrother et al., 2018; Packer et al., 2018; Cobbe et al., 2018; Song et al., 2019; Raileanu & Fergus, 2021). Mitigating disastrous failures of RL-trained agents in practice can prevent many undesirable outcomes (Srinivasan et al., 2020; Choi et al., 2021). Robust MDPs have emerged as a promising solution to mitigate this problem and address the issue of the sensitivity of reinforcement learning to changing environments. By optimizing policies to have large value functions even in the worst-case environmental dynamics, Robust MDPs provide a more stable approach to training agents (Nilim & Ghaoui, 2003).

However, designing algorithms to solve Robust MDPs that converge to robust minima is notably tricky. The first problem is that many objectives in Robust MDPs are nonconvex. For example, maximizing the value function is a nonconvex problem for which optimization is much more difficult. The second problem is that many MDPs have complex interactions between the policy and the environment. Many strategies rely on tuning a policy variable and environment in turn and have specially designed updates to account for these complex interactions. Designing such algorithms and proving their convergence is often tricky; people often take stringent and unideal rectangularity assumptions on possible transitions to make convergence easier to prove. As a result, many different variants of algorithms exist for different settings of Robust MDPs, each needing their own analysis on convergence.

However, the convergence of similar iterative procedures has been studied in settings where the optimization objective is convex. The No-Regret Framework is one of the most powerful frameworks for generating such algorithms that converge to robust minima in convex settings. These frameworks phrase the optimization algorithm as a game between two players, much like the turn-based strategies. However, the benefit of No-Regret Frameworks is that the two players' interactions can be decoupled during analysis, and the convergence to robust minima only depends on the regret of the two players' strategies. This decoupling greatly simplifies convergence analysis. Moreover, the framework is versatile since different choices for each player's strategy create different but simple-to-analyze behaviors. For these reasons, using No-Regret Frameworks has led to the most powerful algorithms for finding robust minima in many convex settings, such as Perceptron algorithms (Wang et al., 2022a) or Electronic Markets (Kalita, 2018). While it would be desirable to use No-Regret frameworks as

is for Robust MDPs, the convergence for no-regret frameworks has been only shown for convex optimization objectives. This optimization objective is often not convex in the case of Robust MDPs.

Here, we make our first contribution. We develop a Nonconvex No-Regret Framework, where policy and environment players play nonconvex online learning updates. We demonstrate that convergence to robust minima in expectation for nonconvex loss functions can still be expressed as the sum of the regrets of the two players' strategies. This convergence is a powerful result that allows the simple generation of many different algorithms to find robust minima in nonconvex settings. While this framework is powerful, the toolbox of nonconvex online learning algorithms needs to be augmented. For the first augmentation, the environmental player in our framework can see the incoming loss function, so nonconvex online learning algorithms that can be used for the environment player must be developed. For this reason, we developed two new algorithms known as Best-Response and Follow the Perturbed Leader Plus (FTPL+) that have improved regret rates with this ability to see the incoming loss function. The second augmentation is that the nonconvex online learning literature often depends on a minimization oracle that can find a global minimizer of the nonconvex loss function. While this is impossible to build in every setting, we provide tools for this minimization oracle for Robust MDPs. Namely, we demonstrate that for both the policy and environmental players, the Value Function, a common objective for Robust MDPs, exhibits gradient dominance. Thus, using Projected Gradient Descent (PGD) will surprisingly suffice as a minimization oracle for both players. This choice has the added benefit of enjoying the scalability of gradient methods. Thus, under different conditions, including simple gradient dominance, smooth MDPs, or strongly gradient-dominated MDPs, we build different algorithms from our framework using different algorithms that take advantage of each setting. With our framework, proving convergence for each algorithm is simple yet powerful. For example, in the most common setting where the objective function is the Value Function, our algorithm achieves the strong convergence rate of $\frac{1}{\sqrt{T}}$ where $T$ is the number of iterations of our algorithm. Moreover, due to the decoupling of the policy and environmental players' behavior, we do not need any rectangularity assumption; instead, we only need the uncertainty set to be convex.

Overall, we provide different algorithms for Robust MDPs with Gradient Dominance, Smooth MDPs, or strongly Gradient-Dominated MDPs and prove robust convergence rates without any rectangularity assumption on the uncertainty set of environments. Alongside these guarantees, we run small experiments on the convergence behavior of our algorithm where the Value Function is the optimization objective in the GridWorld MDP. Our small experiments corroborate this convergence rate in Appendix A.

**Contributions**   Overall, our contributions are as follows.

1. We develop a Nonconvex No-Regret Framework for two-player games where the iterations needed in expectation for convergence to the robust minima is upper bounded by the regret of the two players. We develop two new nonconvex online learning algorithms, Best-Response and FTPL+, that can be used for the environmental player who can look at the incoming loss function in the No-Regret Framework.

2. We provide tools for demonstrating when the objective function of the Robust MDPs is gradient-dominated. Namely, we show that the Value Function is gradient-dominated for both players. Thus, using Follow the Perturbed Leader and Best-Response for the policy player and environmental player with PGD as a minimization oracle in our No-Regret Framework yields an algorithm that solves the Robust MDP scalably in $\frac{1}{\sqrt{T}}$ time with only a convexity assumption on the set of environments. To our knowledge, this is the best convergence rate for MDPs under these assumptions.

3. When the objective function is smooth, such as in Smooth MDPs, we show that using OFTPL and FTPL+ as the strategies for the policy player and the environment player in our No-Regret Framework achieves an even better convergence rate, which takes advantage of the smoothness. Similarly, when the objective function is strongly gradient dominated (we provide a small example), FTPL for the policy player and Best Response for the environment player yields a convergence rate that takes advantage of the strong gradient dominance. To our knowledge, this is the first work to study Robust MDPs with strong gradient dominance.

# 1 RELATED WORKS

**Robust MDPs**   Some of the first algorithms to solve Robust MDPs with guarantees and empirical performance are via transition dynamics set assumptions(Nilim & Ghaoui, 2003; Iyengar, 2005), Robust Policy Iteration (Mankowitz et al., 2019; Tamar et al., 2014; Sinha & Ghate, 2016) or Robust Q-Learning (Wang & Zou, 2021). Regarding robust policy optimization, Mankowitz et al. (2018) proposed robust learning through policy gradient for learning temporally extended action. Mankowitz et al. (2019) used Bellman contractors to optimize the robust objective but did not provide any sub-optimality guarantees, only convergence. Dong et al. (2022) used an online approach similar to ours but used rectangular uncertainty set assumptions and did not utilize any no-regret dynamics. Wang et al. (2023) and Tamar et al. (2014) discussed a policy iteration approach to improving the Average Reward Robust MDP using Bellman Equations. Wang et al. (2022b) uses a similar game framework but does not use no-regret dynamics or sophisticated online-learning algorithms to achieve their guarantees, achieving a worse convergence rate with less robust guarantees.

**No-Regret Learning**   In the context of game no-regret learning, McMahan & Abernethy (2013) used a similar game framework to solve linear optimization. In contrast, Wang et al. (2022a) used this framework for solving binary linear classification. Wang et al. (2021) used these frameworks to solve Fenchel games, phrasing many optimization algorithms. Daskalakis et al. (2017) showed that using a similar framework using online players for training GANs yields strong theoretical and empirical improvements. The classical Syrgkanis et al. (2015) uses No-Regret Dynamics to solve social welfare games.

# 2 PRELIMINARY

## 2.1 NOTATION

We can state that our MDP is a tuple of $(\mathcal{S}, \mathcal{A}, \mathbb{P}_W, R)$ where $\mathcal{S}$ is a set of states, $\mathcal{A}$ is a set of actions, $\mathbb{P}_W$ is a transition dynamics function parameterized by $W$ that maps a state-action-state triple $s, a, s' \in \mathcal{S} \times \mathcal{A} \times \mathcal{S}$ to a probability between 0 and 1, and $R$ is a reward function that maps a state $s$ to a reward $r$. Moreover, we will denote $R_{\max}$ as a constant denoting the largest possible value of $R$. We will assume that both $\mathcal{S}$ and $\mathcal{A}$ are finite sets. We will design a policy $\pi_\theta$ parameterized by a term $\theta \in \mathcal{T}$ where $\mathcal{T}$ is a bounded convex set of vectors of size $d$ for some constant $d$. This policy $\pi_\theta$ maps a state action pair $s, a \in \mathcal{S} \times \mathcal{A}$ to a probability $[0, 1]$. For most of this paper, we will refer to $\pi_\theta$ as $\pi$ where clear. We denote the value function $V$ of a state $s$ under the policy $\pi$ and transition dynamics $\mathbb{P}_W$ as $V_W^\pi(s)$. Here, $W$ parameterizes the transition dynamics belonging to some bounded convex set $\mathcal{W}$. This function is the expected value function of arriving in a state $s$. We will define $\mu \in \Delta(\mathcal{S})$ as some probability distribution over the initial states. Moreover, we will slightly abuse notation call $V_W^\pi(\mu) = \mu^\top V_W^\pi$ where $V_W^\pi$ is the vector of value functions for all states. Moreover, we will define $d_{s_0}^W(s)$ and $d_{s_0}^\pi(s)$ as the probability distribution over the occupancy of states when a policy $\pi$ interacts with an environment of dynamics $W$ given the initial state $s_0$. Formally, this is written as $d_{s_0}^W(s) = d_{s_0}^\pi(s) = (1 - \gamma) \sum_{t=0}^{\infty} \gamma^t \mathbb{P}(s_t = s | s_0, \pi, W)$. We choose to use either $d_{s_0}^W(s)$ and $d_{s_0}^\pi(s)$ based on the context. Moroever, the state visitation distribution under initial state distribution $\mu$ is formally $d_\mu^\pi(s) = \mathop{\mathbb{E}}_{s_0 \sim \mu}[d_{s_0}^\pi(s)]$ and $d_\mu^W(s) = \mathop{\mathbb{E}}_{s_0 \sim \mu}[d_{s_0}^W(s)]$.

## 2.2 ROBUST POLICIES

Our main goal is to create a policy $\pi$ where the expected value function over the initial state distribution is as large as possible. We will first denote how we will define $V$.

**Definition 2.1.** *Given a policy $\pi$ and a transition dynamics $\mathbb{P}_W$, we define the value function recursively as $V_W^\pi(s) = R(s) + \gamma \sum_{a \in \mathcal{A}} \pi(a, s) \sum_{s' \in \mathcal{S}} \mathbb{P}_W(s', a, s) V_W^\pi(s')$. Here $\gamma$ serves as a discount factor and is a positive constant less than 1.*

Within different formulations of Robust MDPs, there may be different objective functions for the robust optimization, called $g(\pi, W)$. One common setup of Robust MDPs is where $g(\pi, W)$ is simply the Value Function given $\pi$ and $W$, specifically $g(\pi, W) = V_W^\pi(\mu)$. Therefore, the Robust MDP problem is simply the task of finding a policy $\pi$ that maximizes the Value Function under worst-case

transition dynamics as in $\pi = \arg\max\limits_{\pi \in \mathcal{T}} \min\limits_{W \in \mathcal{W}} g(\pi, W) = \arg\max\limits_{\pi \in \mathcal{T}} \min\limits_{W \in \mathcal{W}} V_W^\pi(\mu)$. However, in different setups such as Control or Regularized MDPs (Bhandari & Russo, 2022), it may be desirable to find a policy solving a similar robust optimization problem with a different objective function. For example, one may wish to regularize the policy parameter as in $g(\pi_\theta, W) = V_W^{\pi_\theta}(\mu) - \|\theta\|^2$. To generalize the Robust MDP problem to all such possible MDPs, we will denote the optimization problem as the following.

**Definition 2.2.** *The Robust MDP problem is that of finding $\pi$ such that $\pi = \arg\max\limits_{\pi \in \mathcal{T}} \min\limits_{W \in \mathcal{W}} g(\pi, W)$.*

For the rest of this paper, we will treat the case where $g(\pi, W) = V_W^\pi(\mu)$ for simplicity unless explicitly noted. To connect this more clearly to work on repeated games, we will view this as finding a policy $\pi$ that minimizes the suboptimality, i.e., $\arg\max\limits_{\pi^* \in \mathcal{T}} \min\limits_{W \in \mathcal{W}} g(\pi^*, W) - \min\limits_{W \in \mathcal{W}} g(\pi, W)$.

Improving the robustness of the policy can similarly be seen as reducing the difference between our policy's robustness and the best policy's robustness. The second definition intuitively connects to the online learning literature definition of regret. We will similarly use this algorithmic perspective to design a Robust Policy Optimization algorithm that minimizes the suboptimality of the learned policy with the least amount of computational complexity possible.

**Remark 2.1.** *We note that this work does not include details about the differences in optimizing an empirical estimate of the Value Function and the true Value Function. Our contribution lies in the efficient optimization of a deterministic Value Function, dealing with the nonconvex interactions between a policy and the environment. However, this work can possibly be extended to the nondeterministic Value Function setting (Dong et al., 2022; Panaganti & Kalathil, 2020)*

## 3 NO REGRET DYNAMICS

### 3.1 ALGORITHM DETAILS

This framework frames the suboptimality of a policy as a min-max game between the $\pi$ player and the $W$ player. We solve the optimization problem by iteratively choosing a policy and transition dynamics until convergence. Formally, we introduce a policy player, called $OL^\pi$, that chooses a policy $\pi_t$ at time step $t$. Our second player is a $W$-player, called $OL^W$, which will see the

---

**Algorithm 1:** No-Regret RL

**Data:** $T$

**for** $t \in [T]$ **do**
    $OL^\pi$ **chooses a policy:** $\pi_t \leftarrow OL^\pi$;
    $OL^W$ **sees the policy:** $OL^W \leftarrow \pi_t$;
    $OL^W$ **chooses environment:** $W_t \leftarrow OL^W$;
    $OL^\pi$ **sees the environment:** $OL^\pi \leftarrow W_t$;
    $OL^W$ **incur losses:** $OL^W \leftarrow l_t(W_t)$;
    $OL^\pi$ **incur losses:** $OL^\pi \leftarrow h_t(\pi_t)$;
**end**

---

policy $\pi_t$ outputted by the policy player and, then, outputs a transition dynamic $W_t$. The policy player then sees the $W_t$ that was chosen. The policy player incurs a loss $h_t(\pi_t)$, and the environment player then incurs a loss $l_t(W_t)$ corresponding to their decision. They repeatedly play this game until an equilibrium is reached. Our framework can be seen as in Algorithm 1.

This framework is a simple and versatile method of solving many optimization problems. We need only choose the online algorithms that the $\pi$-player and $W$-player employ and the loss function they see. In No-Regret Dynamics, when the loss functions for both players are negative of the others, the algorithm's convergence is simply the convergence for two players. Therefore, if we set $l_t(W_t) = g(W_t, \pi_t)$ and $h_t(\pi_t) = -g(W_t, \pi_t)$, we have Theorem 3.1.

**Theorem 3.1.** *We have the difference between the robustnesses of the chosen policies and any policy $\bar{\pi}$ is upper bounded by the regret of the two players*

$$\frac{1}{T} \min_{W^* \in \mathcal{W}} \sum_{t=0}^{T} g(W^*, \bar{\pi}) - \frac{1}{T} \min_{W^* \in \mathcal{W}} \sum_{t=0}^{T} g(W^*, \pi_t) \leq Reg_W + Reg_\pi.$$

*Here, $Reg_W$ and $Reg_\pi$ are the two average regrets of the two players $OL^W$ and $OL^\pi$.*

As in Theorem 3.1, the convergence of this player is explained by the regret of the two players. This framework is both powerful and intuitive. Say we are in the traditional Robust MDP setting where $g(W, \pi) = V_W^\pi(\mu)$. We can get convergence guarantees for the robustness of our setting by

configuring the loss functions $l_t$ and $h_t$ in the following way. In this setting, the $t$th loss function for the $W$-player is $l_t(W_t) = V_{W_t}^{\pi_t}(\mu)$ and for the $\pi$-player is $h_t(\pi_t) = -V_{W_t}^{\pi_t}(\mu)$. Setting the loss functions according to the above, we similarly get the following convergence guarantee based on the regret of the two players, just like in Theorem 3.1. Therefore, despite the nonconvexity, we have a simple framework for solving the Robust MDP problem. However, we still must choose online learning algorithms for both players and bound their regret to use this framework. Notably, many recent online learning results require the underlying loss functions to be convex or even strongly convex. Regrettably, in the simplest setting where $g(W, \pi) = V_W^{\pi}(\mu)$, neither of these properties are satisfied. Therefore, we must look for efficient online learning algorithms for nonconvex loss functions.

## 4 REGRET MINIMIZATION FOR SMOOTH NONCONVEX LOSS FUNCTIONS

Here, we will develop a toolbox of nonconvex online learning algorithms usable in our framework. First, we will use two algorithms from Suggala & Netrapalli (2019). They present two algorithms that achieve strong convergence in expectation in nonconvex settings: Follow the Perturbed Leader and Optimistic Follow the Perturbed Leader. At each time step, the Follow the Perturbed Leader algorithm generates a random noise vector $\sigma_t$ according to an Exponential distribution with parameter $\eta$. It then chooses $x_t = \mathcal{O}_\alpha \left( \sum_{i=1}^{t-1} f_i - \sigma_i \right)$. Here, $\mathcal{O}_\alpha$ is an Approximate Optimization Oracle that approximately minimizes the received loss function $f_i$ to accuracy $\alpha$. The Optimistic Follow the Perturbed Leader follows a similar procedure, except it chooses $x_t = \mathcal{O}_\alpha \left( g_t + \sum_{i=1}^{t-1} f_i - \sigma_i \right)$ where $g_t$ is some optimistic function. The main dependency of their regret bounds is that there exists an Approximate Optimization Oracle $\mathcal{O}_\alpha$. In this setting, such an oracle takes a nonconvex function $f$ and a noise vector $\sigma$ and returns an approximate minimizer $x^*$ such that $f(x^*) - \langle \sigma, x^* \rangle \leq \left[ \inf_x f(x) - \langle \sigma, x \rangle \right] + \alpha$. Throughout this section, we will assume the presence of such an oracle and discuss how to set this oracle later. They then prove a regret bound for Follow the Perturbed Leader and Optimistic Follow the Perturbed Leader algorithms when they can access such an oracle.

However, for our framework, the environmental player can see the incoming loss function. Neither of the aforementioned methods can account for this. Thus, we will prove that a "Best Response" algorithm achieves an even smaller upper bound regarding regret. The Best Response algorithm assumes knowledge of the coming loss function and returns the value that minimizes the incoming loss function. Formally, the Best Response algorithm outputs $x_t = \arg\min_{x_t} f_t(x_t)$. This is a useful algorithm given that the $W$ player can see the output of the $\pi$ player when making its decision and can greatly reduce its regret. We prove this in the following regret bond.

**Lemma 4.1.** *Suppose we have an incoming sequence of loss functions $f_t$ for $t \in [T]$ with an optimization oracle that can minimize a function to less than $\alpha$ error. The Best Response algorithm satisfies the following regret bound $\frac{1}{T} \sum_{t=1}^{T} f_t(x_t) - \frac{1}{T} \inf_{x \in \mathcal{X}} \sum_{t=1}^{T} f_t(x) \leq \alpha$.*

We will also prove the regret of one more algorithm, Follow the Perturbed Leader Plus. Similar to the Follow the Leader Plus algorithm from Wang et al. (2021), Follow the Perturbed Leader Plus assumes knowledge of the incoming loss function and then outputs the minimizer of the sum of all the seen loss functions minus the noise term. Formally, Follow the Perturbed Leader Plus outputs $x_t$ that satisfies $x_t = \arg\min_{x_t} \sum_{i=1}^{t} f_i(x_t) - \langle \sigma, x_t \rangle$. While this algorithm achieves worse regret than Best Response, it produces more stable outcomes. This will be useful for later extensions. In fact, Follow the Perturbed Leader Plus is equivalent to OFTPL, when the optimistic function $m_t = f_t$ is the true loss function $f_t$. We present the regret of this algorithm here.

**Lemma 4.2.** *Assume access to an optimization oracle that yields solutions with at most error $\alpha$. Given a series of choices by FTPL+ $x_1, \ldots, x_T$ with smooth and Gradient Dominated loss functions $f_1, \ldots, f_t$, the regret in expectation is upper bounded by*

$$\mathbb{E} \left[ \frac{1}{T} \sum_{t=1}^{T} f_t(x_t) - \frac{1}{T} \inf_{x \in \mathcal{X}} \sum_{t=1}^{T} f_t(x) \right] \leq O \left( \frac{dD}{\eta T} + \alpha \right).$$

We provide a summary of these algorithms in the appendix. These two algorithms are suitable choices for the environmental player, and the two algorithms from Suggala & Netrapalli (2019) are suitable choices for the policy player. However, we still have the issue that an Approximation Optimization Oracle is challenging to parameterize. While the objective functions of Robust MDPs do not exhibit convexity, they exhibit *Gradient Dominance* in some settings. This property helps us design a sufficient Approximation Oracle.

## 5 GRADIENT DOMINANCE

Designing a sufficient Approximation Optimization Oracle is a complex problem. However, for Gradient Dominated functions, one can prove convergence guarantees. Notably, we call a function Gradient Dominated if the difference between the function value at a point $x$ and the optimal function value is upper bounded on the order of the function's gradient at the point $x$. We formalize this in the below definition.

**Definition 5.1.** *We say a function $f$ is Gradient Dominated for set $\mathcal{X}$ with constant $\mathcal{K}$ if $f(x) - \min_{x* \in \mathcal{X}} f(x^*) \leq \mathcal{K} \min_{\bar{x} \in \mathcal{X}} \langle \bar{x} - x, \nabla f(x) \rangle$. Here, $\mathcal{K}$ is some constant greater than $0$.*

As noted by Bhandari & Russo (2022), this Gradient Dominated property is relatively common for many different settings, including Quadratic Control or Direct Parameterization. It is useful for proving convergence guarantees for traditional policy gradient methods (Agarwal et al., 2019). For Gradient Dominated smooth functions, one can use projected gradient descent to minimize the function $f$. Namely, Bhandari & Russo (2022) shows that

**Lemma 5.1.** *The below property only holds if $f$ is Gradient Dominated and $\beta \leq \min\left\{\frac{1}{\sup_x \|\nabla f(x)\|_2}, \frac{1}{L}\right\}$. Here, $T_{\mathcal{O}}$ is the number of iterations of Projected Gradient Descent runs. Moreover, the function $c_x$ is used for brevity for $c_\pi$ and $c_W$ later. Also, $D = \max_{x, x' \in \mathcal{X}} \|x - x'\|_2$. Given that $\nabla f$ is $L$-Lipschitz continuous, the sequence $x_{t+1} = \text{Proj}_{\mathcal{X}}(x_t - \beta \nabla f(x_t))$ enjoys the property*

$$f(x_{T_{\mathcal{O}}}) - \inf_{x^*} f(x^*) \leq \sqrt{\frac{2D^2 \mathcal{K}^2 (f(x_0) - \inf_{x^*} f(x^*))}{\beta T_{\mathcal{O}}}} = c_x(T_{\mathcal{O}}, \mathcal{K}).$$

Ideally, we would like to use the straightforward projected gradient descent as our Approximate Optimization Oracle for $OL^\pi$ and $OL^W$. However, this would require the loss functions $l_t$ and $h_t$ to be Gradient Dominated. While not generally true, this is known to hold in many cases. Gradient Dominance is a well-known phenomenon for the $\pi$-player, as seen in Agarwal et al. (2019). We formally list some helpful conditions here.

**Condition 5.1.** *Here, we list the conditions we have.*

1. *The function $\sum_i^t f_i(x) - \langle \sigma, x \rangle$ is Gradient Dominated, enabling the use of FTPL+*

2. *The function $\sum_i^{t-1} f_i(x) - \langle \sigma, x \rangle$ is Gradient Dominated, enabling the use of FTPL.*

3. *The function $\sum_i^{t-1} f_i(x) + f_{t-1}(x) - \langle \sigma, x \rangle$ is Gradient Dominated, enabling the use of OFTPL.*

4. *The function $f_t(x)$ is Gradient Dominated, enabling the use of Best Response.*

We will first provide tools useful for showing when these conditions hold.

### 5.1 TOOLS FOR DEMONSTRATING GRADIENT DOMINANCE

Here, we will provide the tools to show that any of these conditions hold for the objective functions for either the $\pi$ or $W$ players. We have a gradient term within the terms of Gradient Dominance from Definition 5.1. In many cases, the loss function will often contain the Value Function. Therefore, we must know what the gradients of the Value Functions will be for both players. While this is known for the policy player from the Policy Gradient Theorem (Sutton & Barto, 2018), we demonstrate a similar result for the gradient of the Value Function for the $W$-player.

**Lemma 5.2.** *The gradient of the value function $V^W$ with respect to the parameter $W$ is*

$$\nabla_W V^W(s) = \frac{1}{1-\gamma} \sum_{s',a,s} d_\mu^W(s) \mathbb{P}_W(s',a,s) \pi(a|s) \nabla \mathbb{P}_W(s',a,s) V^W(s').$$

Now, we express the suboptimality of a transition dynamics parameter $W$ in terms of the gradient of the Value Function. While this is shown via the Performance Difference Lemma from Kakade & Langford (2002) for the policy player, we need a similar lemma for the transition dynamics. The Performance Difference Lemma relies on the *Advantage function*. We will define an analogous advantage function for the transition dynamics. Intuitively, this Advantage Function is the value of taking state $s'$ over the expected value over all states. Given this, we can provide an analog of the Performance Difference Lemma for the $W$-Player. We provide such a lemma here.

**Lemma 5.3.** *Given two different transition dynamics parameters $W$ and $W'$, we have that $V_W(\mu) - V_{W'}(\mu) = \sum_{s',a,s} d_\mu^W(s) \pi(a|s) \mathbb{P}_W(s'|a,s) A^{W'}(s',a,s)$. Here, we define the $W$-Advantage Function as $A^W(s',a,s) = \gamma V_W(s') + r(s,a) - V_W(s)$.*

We now have ways for the policy and transition dynamics players to analyze both sides of Definition 5.1. We now sufficiently have tools to demonstrate Gradient Dominance in many cases.

## 5.2 Direct Parameterization

We now begin with the direct parameterization case with standard Robust MDPs, where $\mathbb{P}_W(s',a,s) = W_{s',a,s}$ directly parameterizes the transition dynamics, $\pi(s|a) = \theta_{s,a}$, and $g(W,\pi) = V_W^\pi(\mu)$. Moreover, the set of transition dynamics is some convex bounded set, such as a rectangular uncertainty set (Dong et al., 2022). We demonstrate that in this setting, Item 2 holds for the loss function of the policy player, and Item 4 holds for the loss function of the $W$-player.

**Lemma 5.4.** *For any positive noise term $\sigma$, we have that under Direct Parameterization, $\sum_i^{t-1} l_i(\cdot) - \langle \sigma, \cdot \rangle$ and $\sum_i^{t-1} h_i(\cdot) - \langle \sigma, \cdot \rangle$ are both Gradient Dominated for the $W$-player and the $\pi$-player on sets $\mathcal{W}$ and $\mathcal{T}$ with constants $\mathcal{K}_W = \frac{1}{1-\gamma} \left\| \frac{d_\mu^{W^*}}{\mu} \right\|_\infty$ and $\mathcal{K}_\pi = \frac{1}{1-\gamma} \left\| \frac{d_\mu^{\pi^*}}{\mu} \right\|_\infty$ respectively. Therefore, Item 2 hold for both the loss functions for both players.*

Now, we have shown that the loss function for the policy player satisfies Item 2 in this setting. Therefore, we can use Follow the Perturbed Leader for $OL^\pi$. Now, we wish to show Item 4 holds for the loss function of the $W$ player.

**Lemma 5.5.** *We have that the $V_W(\mu)$ is Gradient Dominated with constant $\mathcal{K}_W = \frac{1}{1-\gamma} \left\| \frac{d_\mu^{W^*}}{\mu} \right\|_\infty$ as in for an arbitrary $W \in \mathcal{W}$ and the optimal parameter $W^* \in \mathcal{W}$, we have*

$$V_W(\mu) - V_{W^*}(\mu) \le \mathcal{K}_W \max_{\bar{W} \in \mathcal{W}} \left[ \left(W - \bar{W}\right)^\top \nabla_W V_W(\mu) \right].$$

*Here, we have that Item 4 holds for the $W$-player.*

Now that we have that Item 4 holds for the $W$ player, we know we can use Best Response for the $OL^W$ player. Now, we have that in the Direct Paramaterization setting with standard Robust MDPs; we can use our framework from Algorithm 1 with Follow the Perturbed Leader for $OL^\pi$ and Best Response for $OL^W$ where both use Projected Gradient Descent as their Approximate Optimization Oracle. We can now prove the convergence and robustness of our framework using our proof framework.

## 6 Putting the Bounds Together

We now can achieve bounds for the regrets of either player. Now, if the policy player employs FTPL and the environment player employs Best-Response, when the objective function is the Value Function, we get the following convergence bound.

**Theorem 6.1.** *Assume that the set of possible transition matrices $\mathcal{W}$ is convex. Let $L_\pi$ be the smoothness constant of $h_t$ with respect to the $\ell_1$ norm, $D_\pi = \max_{\pi,\pi' \in \mathcal{T}} \|\pi - \pi'\|_2$, and $d_\pi$ be the dimension of the input for the $\pi$-player. Let $OL^\pi$ use FTPL and $OL^W$ use Best-Response. When $\eta = \frac{1}{L_\pi \sqrt{T d_\pi}}$ the robustness of the set of trained algorithms is for any $\bar{\pi}$*

$$\min_{W^* \in \mathcal{W}} \sum_{t=0}^T V_{W^*}^{\bar{\pi}}(\mu) - \mathbb{E}\left[\min_{W^* \in \mathcal{W}} \sum_{t=0}^T V_{W^*}^{\pi_t}(\mu)\right] \leq \frac{2d_\pi^{\frac{3}{2}} D_\pi L_\pi}{\sqrt{T}} + c_\pi\left(T_\mathcal{O}, \mathcal{K}_\pi\right) + c_W\left(T_\mathcal{O}, \mathcal{K}_W\right).$$

Here, we have shown that in this simple setting, we have that the robustness in expectation is $\mathcal{O}\left(\frac{1}{T^{\frac{1}{2}}} + \frac{1}{T_\mathcal{O}^{\frac{1}{2}}}\right)$ with simple gradient dominance assumptions. However, given some additional properties, it may be possible to improve this bound slightly. We will investigate this in the following sections.

## 7 EXTENSIONS

This robustness works with only the gradient dominance property of each player's loss function. However, with different assumptions, the algorithm's robustness can be improved by utilizing different properties. The two properties we will investigate are smoothness and strong Gradient Dominance.

### 7.1 SMOOTHNESS

As in Agarwal et al. (2019), for the direct parameterization, we have that the Value Function is smooth with respect to the $\pi$-player as in $V_W^\pi(\mu) - V_W^{\pi'}(\mu) = \mathcal{O}\left(\frac{2\gamma R_{\max}|\mathcal{A}|}{(1-\gamma)^2}\|\pi - \pi'\|_2\right)$. Moreover, it is generally difficult to show that the Lipschitz constant for the difference of value functions under different transition dynamics is even smoother. Intuitively, such a function may be smooth in the following manner

**Condition 7.1.** *The difference in value functions between subsequent rounds is smooth with respect to policies such that for all $s \in \mathcal{S}$ and policies $\pi$ and $\pi'$, we have that*

$$[g(W, \pi) - g(W', \pi)] - [g(W, \pi') - g(W', \pi')] \leq \tilde{L}\|W - W'\|_1\|\pi - \pi'\|_1.$$

In general, however, it is difficult to show that such an $\tilde{L}$ is smaller than the value that of $\frac{2\gamma R_{\max}|\mathcal{A}|}{(1-\gamma)^2}$. However, if we are in a setting such that $\tilde{L}$ is small, we can take advantage of this by using Optimistic Follow the Perturbed Leader Plus for the $\pi$-player where the optimistic function is simply the last loss function $g_t = h_{t-1}$. As seen in Lemma D.1, the smoothness of $f_t - g_t$ directly factors into the regret. Therefore, we have that $h_t - h_{t-1}$ is very smooth; we can directly improve such a bound. However, from the formulation of Condition 7.1, we need the outputs from the $W$-player so that $\|W - W'\|_1$ is bounded, which we can demonstrate is the case if the $W$-player uses FTPL+. Now, in the Direct Paramterization setting with the traditional objective function, we also show that the loss functions seen by OFTPL for the policy player and FTPL+ satisfy our Gradient Dominance properties. In the Direct Parameterization case with traditional objective, these conditions hold as shown in Lemma C.1 and Lemma C.2 respectively. Furthermore, in this setting, we have the robustness as such. Indeed, if $\tilde{L}$ is small here, we can improve the convergence and robustness of our trained algorithm by approximately a constant of $\frac{\tilde{L}}{L}$.

**Theorem 7.1.** *Assume that the set of possible transition matrices $\mathcal{W}$ is convex. Let $OL^\pi$ use OFTPL and $OL^W$ use FTPL+. Given that Condition 7.1 holds, we have that the robustness of the algorithm is for any $\bar{\pi}$ when setting $\eta = \sqrt{\frac{20L_\pi(d_w D_W + d_\pi D_\pi)^2}{d_\pi} D_\pi \tilde{L}_2 \alpha T}$,*

$$\min_{W^* \in \mathcal{W}} \sum_{t=0}^T V_{W^*}^{\bar{\pi}}(\mu) - \mathbb{E}\left[\min_{W^* \in \mathcal{W}} \sum_{t=0}^T V_{W^*}^{\pi_t}(\mu)\right] \leq \mathcal{O}\left(\left[\frac{d_\pi \tilde{L}\sqrt{D_\pi(d_w D_w + d_\pi D_\pi)}}{\sqrt{5L_\pi T c_\pi(T_\mathcal{O}, K_\pi)}} + 1\right] c_\pi(T_\mathcal{O}, K_\pi) + c_W\left(T_\mathcal{O}, \mathcal{K}_W\right)\right).$$

## 7.2 STRONG GRADIENT DOMINANCE

Moreover, in specific parameterizations of interest, particular objective functions setup will obey what we call Strongly Gradient Dominance, also known famously as the Polyak-Lojasiewicz condition (Polyak, 1963). This condition helps improve convergence in nonconvex settings, analogous to the strong convexity condition. We formally state such a property in Condition 7.2

**Condition 7.2.** *A function $a(x) = \sum_t l_t(x) - \langle \sigma, x \rangle$ satisfies $\mathcal{K}, \gamma$-strong Gradient Dominance if for any point $x \in \mathcal{X}$,*

$$\min_{x^* \in \mathcal{X}} a(x^*) \geq a(x) - \min_{x' \in \mathcal{X}} \left[ \mathcal{K} \langle x' - x, \nabla a(x) \rangle + \frac{\gamma}{2} \| x' - x \| \right].$$

In general, this is useful for achieving even tighter optimization bounds. Indeed, from Karimi et al. (2016), if we have this, Projected Gradient Descent enjoys better convergence.

**Lemma 7.1.** *Here, $x^* = \underset{x^* \in \mathcal{X}}{\arg \min}\, a_t(x^*)$ is the minimizer of $a_t$. $c_x^s$ is used for brevity. Given $a_t$ is $L$-Lipschitz continuous and is $\mathcal{K}, \gamma$-strongly dominated, we have that using projected gradient descent gets global linear convergence as in*

$$a_t(x_k) - a_t(x^*) \leq \left( 1 - \frac{\gamma}{\mathcal{K}^2 L} \right)^k (a_t(x_0) - a_t(x^*)) = c_x^s(k, \gamma, \mathcal{K}).$$

## 7.3 DIRECT PARAMETERIZATION WITH REGULARIZATION

Say we are in a setting where we want to maximize $V_W^\pi$ and ensure that the policy $\pi$ is regularized according to the $\ell_2$ norm. In this way, we can redefine the loss function to be $g(W, \pi) = V_W^\pi(\mu) - \|\pi\|_2^2$. Again, the sets $\mathcal{W}$ and $\mathcal{T}$ are bounded convex sets in this setting.

**Lemma 7.2.** *We have $g(W, \pi) = V_W^\pi(\mu) - \|\pi\|_2^2$ is $\mathcal{K}_\pi, \frac{T}{2}$-strongly-Gradient Dominated on set $\mathcal{T}$*

In this setting, if the $\pi$ player employs FTPL where its Approximate Optimization Oracle enjoys even better convergence and the $W$-Player employs Best Response, we have the robustness bound as follows. Indeed, given strong Gradient Dominance, we have that the dependence on the complexity for the Optimization Oracle is better for the $\pi$-player, slightly improving the robustness bounds.

**Theorem 7.2.** *Assume that the set of possible transition matrices $\mathcal{W}$ is convex. Let $OL^\pi$ use FTPL and $OL^W$ use Best-Response. Given that Condition 7.2 holds and $\eta = \frac{1}{L_\pi \sqrt{T d_\pi}}$, we have that the robustness of Algorithm 1 is for any $\bar{\pi}$*

$$\min_{W^* \in \mathcal{W}} \sum_{t=0}^{T} V_{W^*}^{\bar{\pi}}(\mu) - \|\bar{\pi}\|^2 - \min_{W^* \in \mathcal{W}} \sum_{t=0}^{T} V_{W^*}^{\pi_t}(\mu) + \|\pi_t\|^2 \leq \frac{2 d_\pi^{\frac{3}{2}} D_\pi L_\pi}{\sqrt{T}} + c_\pi^s \left( T_\mathcal{O}, \mathcal{K}_\pi, \frac{1}{2} \right) + c_W(T_\mathcal{O}, \mathcal{K}_W).$$

# 8 LIMITATIONS AND DISCUSSION

We have designed a Nonconvex No-Regret Framework that has decoupled the convergence for Robust MDP algorithms. With this, we have designed different Robust MDP algorithms for solving different Robust MDPs under standard Gradient Dominance, Strong Gradient Dominance, and Smooth MDPs. The proven convergence results are some of the strongest in the literature, with only a convexity assumption on the set of possible transition matrices. Possible extensions include using this nonconvex No-Regret Framework for other nonconvex problems, such as other nonconvex games, or exploring how different minimization oracles could empirically improve the performance of our algorithms. Another possible avenue could be studying Robust MDPs where the optimization objective obeys the strict saddle property.

**Limitations** However, our work has several limitations. Firstly, we require some Gradient Dominance conditions for the policy and environmental dynamics player. In many settings, the objective function for the Robust MDP does not satisfy this. This assumption does reduce the scope of our work. Moreover, many of our convergence guarantees are only made in expectation, while many other bounds in the literature are made absolutely. Moreover, we do not generate a single policy that achieves strong robustness but, instead, a series of policies that, if used together, obey our convergence guarantees.

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

---

**Algorithm 2:** A set of useful online learning algorithms

---

**Input:** $\eta, \mathcal{O}_\alpha$
**for** $t = 1, \ldots, T$ **do**

$$\text{Sample } \sigma \in \mathbb{R}^d \text{ and } \sigma_i \sim \text{Exp}(\eta) \text{ where } i \in [d]$$

$$\text{FTPL} : x_t = \mathcal{O}_\alpha \left( \sum_{i=1}^{t-1} f_i(x) - \langle \sigma, x \rangle \right)$$

$$\text{OFTPL}[g_t] : x_t = \mathcal{O}_\alpha \left( \sum_{i=1}^{t-1} f_i(x) + g_t(x) - \langle \sigma, x \rangle \right)$$

$$\text{Best-Response} : x_t = \mathcal{O}_\alpha \left( f_t(x) \right)$$

$$\text{FTPL+} : x_t = \mathcal{O}_\alpha \left( \sum_{i=1}^{t} f_i(x) - \langle \sigma, x \rangle \right)$$

**end**

---

## A  EXPERIMENTS

We now turn to verify the algorithm numerically. We will use our algorithm to optimize a policy in the GridWorld MDP (Sutton & Barto, 2018). This setting is a traditional MDP where the world is a grid where the initial state is one corner of the grid. The goal state is the opposite corner of the grid. At each step, the policy can take any of four actions. The next state is sampled respectively from the transition matrix. If the policy lands in the goal state, it receives a reward of $10$, and the MDP is finished. Otherwise, it receives a reward of $-1$. We wish to measure how quickly the robustness of policy is improved through each iteration of our algorithm. As a metric to measure robustness, given a policy, we choose the transition matrix that minimizes the expected reward of the initial state and reports the initial state's expected reward. We do this for every iteration of our algorithm. We will do this over different adversarial transition matrix sets. The sets in question will be

$$\mathcal{T} = \{T \text{ s.t. } \|T - T_0\|_q \leq \gamma\}.$$

Here, $T_0$ is some randomly generated initial transition matrix, $q$ is a hyperparameter affecting the shape of the transition set, and $\gamma$ is the radius of the transition set. We demonstrate the improvement of robustness over several different values of $\gamma$ and $q$. We plot the convergence of our algorithm over $q \in \{1, 2\}$ and $\gamma \in \{.1, .2, .3, .5\}$ in Figure 1.

## B  PROOF OF PRELIMINARY THEOREMS

### B.1  PROOF OF THEOREM 3.1

**Theorem 3.1.** *We have the difference between the robustnesses of the chosen policies and any policy $\bar{\pi}$ is upper bounded by the regret of the two players*

$$\frac{1}{T} \min_{W^* \in \mathcal{W}} \sum_{t=0}^{T} g(W^*, \bar{\pi}) - \frac{1}{T} \min_{W^* \in \mathcal{W}} \sum_{t=0}^{T} g(W^*, \pi_t) \leq Reg_W + Reg_\pi.$$

*Here, $Reg_W$ and $Reg_\pi$ are the two average regrets of the two players $OL^W$ and $OL^\pi$.*

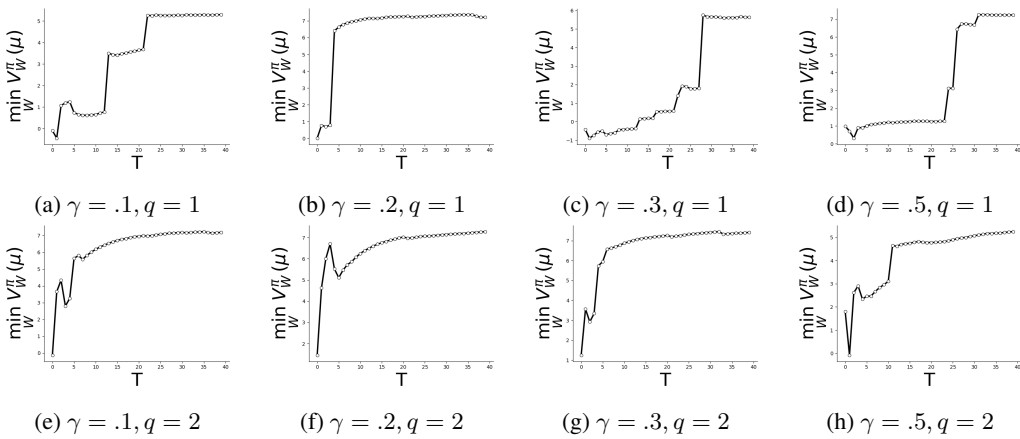

Figure 1: We plot the convergence of our algorithm over many different transition matrix uncertainty set shapes. We see that over all shapes, our algorithm converges in roughly the predicted $\frac{1}{\sqrt{T}}$ rate predicted by our results.

*Proof.* By definition, the regret of the $W$-player is equivalent to

$$\text{Reg}_W = \sum_{t=0}^{T} l_t(W_t) - \min_{W^*} \sum_{t=0}^{T} l_t(W^*)$$

$$= \sum_{t=0}^{T} g(W_t, \pi_t) - \min_{W^*} \sum_{t=0}^{T} g(W^*, \pi_t)$$

Similarly, for the $\pi$ player, we have that

$$\text{Reg}_\pi = \sum_{t=0}^{T} l_t(\pi_t) - \min_{\pi^*} \sum_{t=0}^{T} l_t(\pi^*)$$

$$= \max_{\pi^*} \sum_{t=0}^{T} g(W_t, \pi^*) - \sum_{t=0}^{T} g(W_t, \pi_t) \tag{1}$$

Therefore, we can upper bound the sum of objective functions throughout our training process as

$$\sum_{t=0}^{T} g(W_t, \pi_t) = \text{Reg}_W + \min_{W^*} \sum_{t=0}^{T} g(W^*, \pi_t).$$

We similarly lower bound the sum of value functions throughout our training process as

$$\sum_{t=0}^{T} g(W_t, \pi_t) = \max_{\pi^*} \sum_{t=0}^{T} g(W_t, \pi^*) - \text{Reg}_\pi$$

$$\geq \sum_{t=0}^{T} g(W_t, \bar{\pi}) - \text{Reg}_\pi$$

$$\geq \min_{W^*} \sum_{t=0}^{T} g(W^*, \bar{\pi}) - \text{Reg}_\pi \tag{2}$$

Combining Equation (1) and Equation (2), we have our desired statement

$$\min_{W^*} \sum_{t=0}^{T} g(W^*, \bar{\pi}) - \min_{W^*} \sum_{t=0}^{T} g(W^*, \pi_t) \leq \text{Reg}_W + \text{Reg}_\pi.$$

$\square$

## C    PROOFS OF GRADIENT DOMINANCE

### C.1    PROOF OF LEMMA 5.2

**Lemma 5.2.** *The gradient of the value function $V^W$ with respect to the parameter $W$ is*

$$\nabla_W V^W(s) = \frac{1}{1-\gamma} \sum_{s',a,s} d_\mu^W(s) \mathbb{P}_W(s',a,s) \pi(a|s) \nabla \mathbb{P}_W(s',a,s) V^W(s').$$

*Proof.* We wish to calculate the gradient of the value function $V^W$ with respect to $\mathbb{P}_W(s',a,s)$. We have that

$$\begin{aligned}
\nabla_W V^W(s) &= \nabla_W \left( \sum_a \pi(a|s) Q_\pi(s,a) \right) \\
&= \sum_a \pi(a|s) \nabla_W Q_\pi(s,a) \\
&= \sum_a \pi(a|s) \nabla_W \left[ R(s,a) + \gamma \sum_{s'} \mathbb{P}_W(s',a,s) V^\pi(s') \right] \\
&= \sum_a \gamma \pi(a|s) \sum_{s'} [\nabla_W \mathbb{P}_W(s',a,s) V^\pi(s') + \mathbb{P}_W(s',a,s) \nabla_W V^\pi(s')] \\
&= \sum_{a,s'} \gamma \pi(a|s) \mathbb{P}_W(s',a,s) \nabla_W V^\pi(s') + \sum_{s',a} \pi(a|s) \gamma \nabla_W \mathbb{P}_W(s',a,s) V^\pi(s')
\end{aligned}$$

Here, the third inequality comes from the definition of the $Q$ function. Unrolling this makes it such that we have

$$\begin{aligned}
\nabla_W V^W(\mu) &= \sum_{t=0}^{\infty} \sum_{s',a,s} Pr(s_t = s|\mu) \gamma^t \pi(a|s) \nabla \mathbb{P}_W(s',a,s) V^W(s') \\
&= \frac{1}{1-\gamma} \sum_{s',a,s} d_\mu^W(s) \pi(a|s) \nabla \mathbb{P}_W(s',a,s) V^W(s')
\end{aligned}$$

We have now arrived at our desired quantity.                                            $\square$

### C.2    PROOF OF LEMMA 5.3

**Lemma 5.3.** *Given two different transition dynamics parameters $W$ and $W'$, we have that $V_W(\mu) - V_{W'}(\mu) = \sum_{s',a,s} d_\mu^W(s) \pi(a|s) \mathbb{P}_W(s'|a,s) A^{W'}(s',a,s)$. Here, we define the W-Advantage Function as $A^W(s',a,s) = \gamma V_W(s') + r(s,a) - V_W(s)$.*

*Proof.* This proof follows mainly from the proof of the Performance Difference Lemma from Kakade & Langford (2002).

$$\begin{aligned}
V_W(\mu) - V_{W'}(\mu) &= \mathbb{E}_{\mathbb{P}_W,\pi} \sum_{t=0}^{\infty} \gamma^t r(s_t,a_t) - V^{W'}(\mu) \\
&= \mathbb{E}_{\mathbb{P}_W,\pi} \left[ \sum_{t=0}^{\infty} \gamma^t r(s_t,a_t) + \gamma^t V_{W'}(s_t) - \gamma^t V_{W'}(s_t) \right] - V^{W'}(\mu) \\
&= \mathbb{E}_{\mathbb{P}_W,\pi} \left[ \sum_{t=0}^{\infty} \gamma^t r(s_t,a_t) + \gamma^{t+1} V_{W'}(s_{t+1}) - \gamma^t V_{W'}(s_t) \right] \\
&= \mathbb{E}_{\mathbb{P}_W,\pi} \left[ \sum_{t=0}^{\infty} \gamma^t A^{W'}(s_{t+1},a_t,s_t) \right] \\
&= \frac{1}{1-\gamma} \sum_{s',a,s} \left[ \gamma^t d_\mu^W(s) \pi(a|s) \mathbb{P}_W(s'|a,s) A^{W'}(s',a_t,s) \right]
\end{aligned}$$

Here, the fourth equality comes from our definition of the Advantage function for the $W$-player. This concludes the proof. $\qquad\square$

### C.3 PROOF OF LEMMA 5.5

**Lemma 5.5.** *We have that the $V_W(\mu)$ is Gradient Dominated with constant $\mathcal{K}_W = \frac{1}{1-\gamma} \left\| \frac{d_\mu^{W^*}}{\mu} \right\|_\infty$ as in for an arbitrary $W \in \mathcal{W}$ and the optimal parameter $W^* \in \mathcal{W}$, we have*

$$V_W(\mu) - V_{W^*}(\mu) \leq \mathcal{K}_W \max_{\bar{W} \in \mathcal{W}} \left[ (W - \bar{W})^\top \nabla_W V_W(\mu) \right].$$

*Here, we have that Item 4 holds for the $W$-player.*

*Proof.* We can use Lemma 5.3 to prove this. We will lower bound the difference between the optimal $V_{W^*}(\mu)$ and the $V_W(\mu)$. In order to prove Gradient Dominance, we need that $V_W(\mu) - V_{W^*}(\mu)$ to be upper-bounded. We will equivalently lower bound the negative of this. We have

$$V_W(\mu) - V_{W^*}(\mu) = \frac{-1}{1-\gamma} \sum_{s',a,s} \left[ \gamma^t d_\mu^{W^*}(s) \pi(a|s) \mathbb{P}_{W^*}(s'|a,s) A^W(s',a,s) \right]$$

$$\leq \frac{-1}{1-\gamma} \sum_{s',a,s} \left[ \gamma^t d_\mu^{W^*}(s) \pi(a|s) \min_{s'} \left( A^W(s',a,s) \right) \right]$$

$$\leq \left( \max_s \frac{d_\mu^{W^*}(s)}{d_\mu^W(s)} \right) \frac{-1}{1-\gamma} \sum_{s',a,s} \left[ \gamma^t d_\mu^W(s) \pi(a|s) \min_{s'} \left( A^W(s',a,s) \right) \right]$$

Here, the first inequality comes from seeing that the value $\mathbb{P}_{W^*}(s'|a,s) A^W(s',a,s)$ is minimized when $\mathbb{P}_{W^*}$ puts the most weight on the state minimizing the advantage function. Looking only at that last term, we can bound it in the following manner

$$\frac{-1}{1-\gamma} \sum_{s',a,s} \left[ \gamma^t d_\mu^W(s) \pi(a|s) \min_{s'} \left( A^W(s',a,s) \right) \right]$$

$$= \frac{-1}{1-\gamma} \min_{\bar{W}} \sum_{s',a,s} \left[ \gamma^t d_\mu^W(s) \pi(a|s) \mathbb{P}_{\bar{W}}(s',a,s) \left( A^W(s',a,s) \right) \right]$$

$$= \frac{-1}{1-\gamma} \min_{\bar{W}} \sum_{s',a,s} \left[ \gamma^t d_\mu^W(s) \pi(a|s) \left( \mathbb{P}_{\bar{W}}(s',a,s) - \mathbb{P}_W(s',a,s) \right) \left( A^W(s',a,s) \right) \right]$$

$$= \frac{-1}{1-\gamma} \min_{\bar{W}} \sum_{s',a,s} \left[ \gamma^t d_\mu^W(s) \pi(a|s) \left( \mathbb{P}_{\bar{W}}(s',a,s) - \mathbb{P}_W(s',a,s) \right) \left( V_W(s') \right) \right]$$

$$= -\min_{\bar{W}} \left[ (\bar{W} - W)^\top \nabla_{W'} V_W(\mu) \right]$$

Here, the first equality comes from seeing that the value $\mathbb{P}_{W^*}(s'|a,s) A^W(s',a,s)$ is minimized when $\mathbb{P}_{W^*}$ puts the most weight on the state minimizing the advantage function. The second equality comes from the fact that the $\sum_{s'} \mathbb{P}_W(s',a,s) A^W(s',a,s) = 0$. The third inequality comes from the definition of the $W$-player advantage function. Finally, the fourth equality comes from Lemma 5.2. Combining these yield

$$V_W(\mu) - V_{W^*}(\mu) \leq - \left\| \frac{d_\mu^{W^*}}{d_\mu^W} \right\|_\infty \min_{\bar{W}} \left[ (\bar{W} - W)^\top \nabla_{W'} V_W(\mu) \right]$$

$$\leq \frac{-1}{1-\gamma} \left\| \frac{d_\mu^W}{\mu} \right\|_\infty \min_{\bar{W}} \left[ (\bar{W} - W)^\top \nabla_{W'} V_W(\mu) \right]$$

The last inequality comes from the fact that $d_\mu^{W^*}(s) \geq (1-\gamma)\mu(s)$ by definition. Here, flipping this, we have

$$V_W(\mu) - V_{W^*}(\mu) \leq \frac{-1}{1-\gamma} \left\| \frac{d_\mu^{W^*}}{\mu} \right\|_\infty \min_{\bar{W}} \left[ \left( \bar{W} - W \right)^\top \nabla_{W'} V_W(\mu) \right]$$

$$\leq \frac{1}{1-\gamma} \left\| \frac{d_\mu^{W^*}}{\mu} \right\|_\infty \max_{\bar{W}} \left[ \left( W - \bar{W} \right)^\top \nabla_{W'} V_W(\mu) \right]$$

This is a satisfying definition of Gradient Dominance. □

## C.4 PROOF OF LEMMA 5.4

**Lemma 5.4.** *For any positive noise term $\sigma$, we have that under Direct Parameterization, $\sum_i^{t-1} l_i(\cdot) - \langle \sigma, \cdot \rangle$ and $\sum_i^{t-1} h_i(\cdot) - \langle \sigma, \cdot \rangle$ are both Gradient Dominated for the $W$-player and the $\pi$-player on sets $\mathcal{W}$ and $\mathcal{T}$ with constants $\mathcal{K}_W = \frac{1}{1-\gamma} \left\| \frac{d_\mu^{W^*}}{\mu} \right\|_\infty$ and $\mathcal{K}_\pi = \frac{1}{1-\gamma} \left\| \frac{d_\mu^{\pi^*}}{\mu} \right\|_\infty$ respectively. Therefore, Item 2 hold for both the loss functions for both players.*

*Proof.* We can use Lemma 5.3 to prove both. We start with the $W$ player.

$$\sum_i^{t-1} V_W^{\pi_i}(\mu) - V_{W^*}^{\pi_i}(\mu) - \langle \sigma, W - W^* \rangle =$$

$$\sum_i^{t-1} \frac{-1}{1-\gamma} \sum_{s',a,s} \left[ d_\mu^{W^*}(s) \pi_i(a|s) \mathbb{P}_{W^*}(s'|a,s) A^W(s',a,s) \right] - \langle \sigma, W^* - W \rangle$$

$$\leq \left( \max_s \frac{d_\mu^{W^*}(s)}{d_\mu^W(s)} \right) \left[ \sum_i^{t-1} \frac{-1}{1-\gamma} \sum_{s',a,s} \left[ d_\mu^W(s) \pi_i(a|s) \mathbb{P}_{W^*}(s'|a,s) \left( A^W(s',a,s) \right) \right] \right.$$

$$\left. - \langle \sigma, W^* - W \rangle \right]$$

$$\leq \left( \max_s \frac{d_\mu^{W^*}(s)}{d_\mu^W(s)} \right) \max_{\bar{W}} \left[ \sum_i^{t-1} \frac{-1}{1-\gamma} \sum_{s',a,s} \left[ d_\mu^{W^*}(s) \pi_i(a|s) \mathbb{P}_{\bar{W}}(s'|a,s) \left( A^W(s',a,s) \right) \right] \right.$$

$$\left. - \langle \sigma, \bar{W} - W \rangle \right]$$

Here, the first inequality comes from the fact that the maximum of the interior of the RHS is always nonnegative, and the final inequality comes from using the minimizing transition dynamics $\bar{W}$.

Looking at the inside term, we have that

$$\max_{\bar{W}} \left[ \frac{-1}{1-\gamma} \sum_{s',a,s} \sum_i \left[ d_\mu^W(s)\pi_i(a|s)\, \mathbb{P}_{\bar{W}}(s',a,s)\left(A^W(s',a,s)\right)\right] - \langle \sigma, \bar{W} - W \rangle \right]$$

$$= \max_{\bar{W}} \left[ \frac{-1}{1-\gamma} \sum_{s',a,s} \sum_i \left[ d_\mu^W(s)\pi_i(a|s)\, \left(\mathbb{P}_{\bar{W}}(s',a,s) - \mathbb{P}_W(s',a,s)\right)\left(A^W(s',a,s)\right) \right] \right.$$

$$\left. - \langle \sigma, \bar{W} - W \rangle \right]$$

$$= \max_{\bar{W}} \left[ \frac{-1}{1-\gamma} \sum_{s',a,s} \sum_i \left[ d_\mu^W(s)\pi_i(a|s)\, \left(\mathbb{P}_{\bar{W}}(s',a,s) - \mathbb{P}_W(s',a,s)\right)\left(V_W(s')\right)\right] \right.$$

$$\left. - \langle \sigma, \bar{W} - W \rangle \right]$$

$$= \max_{\bar{W}} \left[ \left(W - \bar{W}\right)^\top \nabla_W \left[ \sum_i V_W^{\pi_i}(\mu) - \langle \sigma, W \rangle \right] \right]$$

The first equality comes from the fact that the $\sum_{s'} \mathbb{P}_W(s',a,s)A^W(s',a,s) = 0$. The second inequality comes from the definition of the $W$-player advantage function. Finally, the third equality comes from Lemma 5.2. Combining these, we have that

$$\sum_i^{t-1} V_W^{\pi_i}(\mu) - V_{W^*}^{\pi_i}(\mu) - \langle \sigma, W - W^* \rangle \le$$

$$\left\| \frac{d_\mu^{W^*}}{\mu} \right\|_\infty \frac{1}{1-\gamma} \max_{\bar{W}} \left[ \left(W - \bar{W}\right)^\top \nabla_W \left[ \sum_i V_W^{\pi_i}(\mu) - \langle \sigma, W \rangle \right] \right]$$

Moreover, we use the fact that $d_\mu^W(s) \ge (1-\gamma)\mu(s)$ by definition. We now do this for the $\pi$-player. By the Performance Difference Lemma,

$$\sum_i^{t-1} V_{W_i}^{\pi^*} - V_{W_i}^{\pi} - \langle \sigma, \pi^* - \pi \rangle = \frac{1}{1-\gamma} \sum_i^{t-1} \sum_{s,a} d_\mu^{\pi^*}(s)\pi^*(a,s)A^\pi(s,a) - \langle \sigma, \pi^* - \pi \rangle$$

$$\le \frac{1}{1-\gamma} \max_{\bar{\pi}} \left[ \sum_i^{t-1} \sum_{s,a} d_\mu^{\pi^*}(s)\bar{\pi}(s,a)A^\pi(s,a) - \langle \sigma, \bar{\pi} - \pi \rangle \right]$$

$$\le \left\| \frac{d_\mu^{\pi^*}}{d_\mu^\pi} \right\|_\infty \max_{\bar{\pi}} \left[ \frac{1}{1-\gamma} \sum_i^{t-1} \sum_{s,a} d_\mu^\pi(s)\bar{\pi}(s,a)A^\pi(s,a) - \langle \sigma, \bar{\pi} - \pi \rangle \right]$$

$$= \left\| \frac{d_\mu^{\pi^*}}{d_\mu^\pi} \right\|_\infty \max_{\bar{\pi}} \left[ \frac{1}{1-\gamma} \sum_i^{t-1} \sum_{s,a} d_\mu^\pi(s)(\bar{\pi}(s,a) - \pi(a,s))A^\pi(s,a) - \langle \sigma, \bar{\pi} - \pi \rangle \right]$$

$$= \left\| \frac{d_\mu^{\pi^*}}{d_\mu^\pi} \right\|_\infty \max_{\bar{\pi}} \left[ \frac{1}{1-\gamma} \sum_i^{t-1} \sum_{s,a} d_\mu^\pi(s)(\bar{\pi}(s,a) - \pi(a,s))Q^\pi(s,a) - \langle \sigma, \bar{\pi} - \pi \rangle \right]$$

$$\le \frac{1}{1-\gamma} \left\| \frac{d_\mu^{\pi^*}}{\mu} \right\|_\infty \max_{\bar{\pi}}(\bar{\pi} - \pi)^\top \nabla_\pi \left( \sum_i^{t-1} V_{W_i}^\pi - \langle \sigma, \pi \rangle \right)$$

Here, the first equality comes from the Performance Difference Lemma, the second equality comes from the fact that $\sum_a \pi(a,s)A^\pi(s,a) = 0$, the third equality comes from the definition of the Advantage Function for the $\pi$-player, and the final inequality comes from the both the Policy Gradient Theorem and the fact that $d_\mu^\pi(s) \ge (1-\gamma)\mu(s)$. We now have proven both claims of our lemma. $\square$

**Lemma C.1.** *The $\pi$-player enjoys Item 3, i.e. $\sum_i^{t-1} h_i(\cdot) + h_{t-1}(\cdot) - \sigma$ is gradient-dominated with constant $\frac{1}{1-\gamma} \left\| \frac{d_\mu^{\pi^*}}{\mu} \right\|_\infty$.*

*Proof.* For simplicity, we will call $\sum_j h_j(\cdot) := \sum_i^{t-1} h_i(\cdot) + h_{t-1}(\cdot)$ where $j$ indexes over the set $\{h_1, \ldots, h_{t-2}, h_{t-1}, h_{t-1}\}$. With this, we can follow through with our proof. By the performance difference lemma,

$$\sum_j V_{W_i}^{\pi^*} - V_{W_i}^{\pi} - \langle \sigma, \pi^* - \pi \rangle$$

$$= \frac{1}{1-\gamma} \sum_j \sum_{s,a} d_\mu^{\pi^*}(s) \pi^*(a,s) A^\pi(s,a) - \langle \sigma, \pi^* - \pi \rangle$$

$$\leq \frac{1}{1-\gamma} \max_{\bar\pi} \sum_j \sum_{s,a} d_\mu^{\pi^*}(s) \bar\pi(s,a) A^\pi(s,a) - \langle \sigma, \pi^* - \pi \rangle$$

$$\leq \left\| \frac{d_\mu^{\pi^*}}{d_\mu^\pi} \right\|_\infty \max_{\bar\pi} \left[ \frac{1}{1-\gamma} \sum_j \sum_{s,a} d_\mu^\pi(s) \bar\pi(s,a) A^\pi(s,a) - \langle \sigma, \bar\pi - \pi \rangle \right]$$

$$= \left\| \frac{d_\mu^{\pi^*}}{d_\mu^\pi} \right\|_\infty \max_{\bar\pi} \left[ \frac{1}{1-\gamma} \sum_j \sum_{s,a} d_\mu^\pi(s) (\bar\pi(s,a) - \pi(a,s)) A^\pi(s,a) - \langle \sigma, \bar\pi - \pi \rangle \right]$$

$$= \left\| \frac{d_\mu^{\pi^*}}{d_\mu^\pi} \right\|_\infty \max_{\bar\pi} \left[ \frac{1}{1-\gamma} \sum_j \sum_{s,a} d_\mu^\pi(s) (\bar\pi(s,a) - \pi(a,s)) Q^\pi(s,a) - \langle \sigma, \bar\pi - \pi \rangle \right]$$

$$\leq \frac{1}{1-\gamma} \left\| \frac{d_\mu^{\pi^*}}{\mu} \right\|_\infty \max_{\bar\pi} (\bar\pi - \pi)^\top \nabla_\pi \left( \sum_j V_{W_i}^\pi - \langle \sigma, \pi \rangle \right)$$

Here, the first equality comes from the Performance Difference Lemma, the second equality comes from the fact that $\sum_a \pi(a,s) A^\pi(s,a) = 0$, the third equality comes from the definition of the Advantage Function for the $\pi$-player, and the final inequality comes from the both the Policy Gradient Theorem and the fact that $d_\mu^\pi(s) \geq (1-\gamma)\mu(s)$. $\qquad\square$

**Lemma C.2.** *Item 1 is satisfied for the $W$-player, i.e. $\sum_i^t l_i - \sigma$ is Gradient Dominated.*

*Proof.* We can use Lemma 5.3 to prove both. We start with the $W$ player.

$$\sum_i^t V_W^{\pi_i}(\mu) - V_{W^*}^{\pi_i}(\mu) - \langle \sigma, W - W^* \rangle =$$

$$\sum_i^t \frac{-1}{1-\gamma} \sum_{s',a,s} \left[ d_\mu^{W^*}(s) \pi_i(a|s) \mathbb{P}_{W^*}(s'|a,s) A^W(s',a,s) \right] - \langle \sigma, W^* - W \rangle$$

$$\leq \left\| \frac{d_\mu^{W^*}(s)}{d_\mu^W(s)} \right\|_\infty \left[ \sum_i^t \frac{-1}{1-\gamma} \sum_{s',a,s} \left[ d_\mu^W(s) \pi_i(a|s) \, \mathbb{P}_{W^*}(s'|a,s) \left( A^W(s',a,s) \right) \right] \right.$$
$$\left. - \langle \sigma, W^* - W \rangle \right]$$

$$\leq \left\| \frac{d_\mu^{W^*}(s)}{d_\mu^W(s)} \right\|_\infty \left[ \max_{\bar W} \sum_i^t \frac{-1}{1-\gamma} \sum_{s',a,s} \left[ d_\mu^{W^*}(s) \pi_i(a|s) \mathbb{P}_{\bar W}(s'|a,s) \left( A^W(s',a,s) \right) \right] \right.$$
$$\left. - \langle \sigma, \bar W - W \rangle \right]$$

Here, the final inequality comes from using the minimizing transition dynamics $\bar{W}$. Looking at the inside term, we have that

$$
\max_{\bar{W}} \left[ \frac{-1}{1-\gamma} \sum_{s',a,s} \sum_i \left[ d_\mu^W(s) \pi_i(a|s) \, \mathbb{P}_{\bar{W}}(s',a,s) \left( A^W(s',a,s) \right) \right] - \langle \sigma, \bar{W} - W \rangle \right]
$$

$$
= \max_{\bar{W}} \left[ \frac{-1}{1-\gamma} \sum_{s',a,s} \sum_i \left[ d_\mu^W(s) \pi_i(a|s) \, \left( \mathbb{P}_{\bar{W}}(s',a,s) - \mathbb{P}_W(s',a,s) \right) \left( A^W(s',a,s) \right) \right] \right.
$$

$$
\left. - \langle \sigma, \bar{W} - W \rangle \right]
$$

$$
= \max_{\bar{W}} \left[ \frac{-1}{1-\gamma} \sum_{s',a,s} \sum_i \left[ d_\mu^W(s) \pi_i(a|s) \, \left( \mathbb{P}_{\bar{W}}(s',a,s) - \mathbb{P}_W(s',a,s) \right) \left( V_W(s') \right) \right] \right.
$$

$$
\left. - \langle \sigma, \bar{W} - W \rangle \right]
$$

$$
= \max_{\bar{W}} \left[ \left( W - \bar{W} \right)^\top \nabla_W \left[ \sum_i V_W^{\pi_i}(\mu) - \langle \sigma, W \rangle \right] \right]
$$

The first equality comes from the fact that the $\sum_{s'} \mathbb{P}_W(s',a,s) A^W(s',a,s) = 0$. The second inequality comes from the definition of the $W$-player advantage function. Finally, the third equality comes from Lemma 5.2. Combining these, we have that

$$
\sum_i^t V_W^{\pi_i}(\mu) - V_{W^*}^{\pi_i}(\mu) - \langle \sigma, W - W^* \rangle \leq
$$

$$
\frac{1}{1-\gamma} \left\| \frac{d_\mu^{W^*}}{\mu} \right\|_\infty \max_{\bar{W}} \left[ \left( W - \bar{W} \right)^\top \nabla_W \left[ \sum_i V_W^{\pi_i}(\mu) - \langle \sigma, W \rangle \right] \right]
$$

Moreover, we use the fact that $d_\mu^W(s) \geq (1-\gamma)\mu(s)$ by definition. $\qquad\square$

# D   PROOFS FOR CONVERGENCE

Here, we detail a lemma on the convergence of FTPL and OFTPL as proven in Suggala & Netrapalli (2019).

**Lemma D.1.** *Let $D$ be the $\ell_\infty$ diameter of the space $\mathcal{X}$. Suppose the losses encountered by the learner are $L$-Lipschitz w.r.t $\ell_1$ norm. Moreover, suppose the optimization oracle used has error $\alpha$. For any fixed $\eta$, the predictions of Follow the Perturbed Leader satisfy the following regret bound. Here, $d$ is the dimension of the noise vector.*

$$
\mathbb{E} \left[ \frac{1}{T} \sum_{t=1}^T f_t(x_t) - \frac{1}{T} \inf_{x \in \mathcal{X}} \sum_{t=1}^T f_t(x) \right] \leq O \left( \eta d^2 D L^2 + \frac{dD}{\eta T} + \alpha \right).
$$

*For OFTPL, suppose our guess $g_t$ is such that $g_t - f_t$ is $L_t$-Lipschitz w.r.t $\ell_1$ norm, for all $t \in [T]$. The predictions of Optimistic Follow the Perturbed Leader satisfy the following regret bound.*

$$
\mathbb{E} \left[ \frac{1}{T} \sum_{t=1}^T f_t(x_t) - \frac{1}{T} \inf_{x \in \mathcal{X}} \sum_{t=1}^T f_t(x) \right] \leq O \left( \eta d^2 D \sum_{t=1}^T \frac{L_t^2}{T} + \frac{dD}{\eta T} + \alpha \right).
$$

## D.1   PROOF OF LEMMA 4.1

**Lemma 4.1.** *Suppose we have an incoming sequence of loss functions $f_t$ for $t \in [T]$ with an optimization oracle that can minimize a function to less than $\alpha$ error. The Best Response algorithm satisfies the following regret bound $\frac{1}{T} \sum_{t=1}^T f_t(x_t) - \frac{1}{T} \inf_{x \in \mathcal{X}} \sum_{t=1}^T f_t(x) \leq \alpha$.*

*Proof.* The regret term is defined as $\frac{1}{T}\sum_{t=1}^{T} f_t(x_t) - \frac{1}{T}\inf_{x\in\mathcal{X}}\sum_{t=1}^{T} f_t(x)$ where $x_t$ are the choices taken by the BestResponse algorithm. For an arbitrary time step $t$, we have that

$$f_t(x_t) - f_t(x) \leq \min_{x^*} f_t(x^*) + \alpha - f_t(x)$$
$$\leq \alpha$$

where the first inequality comes from the fact that an optimization oracle is used to calculate $x_t$ and has error upper bounded by $\alpha$. Therefore, we have that

$$\frac{1}{T}\sum_{t=1}^{T} f_t(x_t) - \frac{1}{T}\inf_{x\in\mathcal{X}}\sum_{t=1}^{T} f_t(x) \leq \alpha.$$

$\square$

# E  PROOFS FOR EXTENSION SECTION

## E.1  PROOF OF LEMMA E.1

**Lemma E.1.** *Given a series of choices by FTPL+ $x_1,\ldots,x_T$ with smooth and Gradient Dominated loss functions $f_1,\ldots,f_t$ and noise sampled $\sigma \sim Exp(\eta)$, the stability in choices is bounded in expectation by*

$$\mathbb{E}(\|x_{t+1} - x_t\|_1) \leq 125\eta L d^2 D + \frac{\alpha}{20L}$$

To prove this, we will first prove two properties of monotonicity of the loss function on an input of noise $\sigma$. Much of this proof structure is inspired by Suggala & Netrapalli (2019).

**Lemma E.2.** *Let $x_t(\sigma)$ be the solution chosen by FTPL+ under noise sigma. Let $\sigma' = \sigma + ce_i$ for some positive constant c, then we have that*

$$x_{t,i}(\sigma') \geq x_{t,i}(\sigma) - \frac{2\alpha}{c}.$$

*Proof.* Given that the approximate optimality of $x_t(\sigma)$, we have that

$$\sum_{i=1}^{t-1} f_i(x_t(\sigma)) + m_t(x_t(\sigma)) - \langle\sigma, x_t(\sigma)\rangle \tag{3}$$

$$\leq \sum_{i=1}^{t-1} f_i(x_t(\sigma')) + m_t(x_t(\sigma')) - \langle\sigma, x_t(\sigma')\rangle + \alpha$$

$$= \sum_{i=1}^{t-1} f_i(x_t(\sigma')) + m_t(x_t(\sigma')) - \langle\sigma', x_t(\sigma')\rangle + \langle\sigma' - \sigma, x_t(\sigma')\rangle + \alpha$$

$$\leq \sum_{i=1}^{t-1} f_i(x_t(\sigma)) + m_t(x_t(\sigma)) - \langle\sigma', x_t(\sigma)\rangle + \langle\sigma' - \sigma, x_t(\sigma')\rangle + 2\alpha$$

$$= \sum_{i=1}^{t-1} f_i(x_t(\sigma)) + m_t(x_t(\sigma)) - \langle\sigma, x_t(\sigma)\rangle + \langle\sigma' - \sigma, x_t(\sigma') - x_t(\sigma)\rangle + 2\alpha \tag{4}$$

Here, the first equality comes from the fact that $x_t(\sigma)$ is an approximate minimizer for the loss function, and the second inequality comes from the fact that $x_t(\sigma')$ minimizes the loss function with noise set to $\sigma'$ by definition. Combining Equation (3) and Equation (4), we have that

$$0 \leq \langle\sigma' - \sigma, x_t(\sigma') - x_t(\sigma)\rangle + 2\alpha$$
$$\leq c(x_{(t,i)}(\sigma') - x_{(t,i)}(\sigma)) + 2\alpha$$

We, therefore, get that $x_{(t,i)}(\sigma') \geq x_{(t,i)}(\sigma) - \frac{2\alpha}{c}$.

$\square$

Moreover, we have that the difference between predictions made by our algorithm at sequential timesteps is close.

**Lemma E.3.** *If $\|x_t(\sigma) - x_{t+1}(\sigma)\|_1 \leq 10d|x_{t,i}(\sigma) - x_{t+1,i}(\sigma)|$ and $\sigma' = 100Lde_i + \sigma$, we have that*

$$\min\left(x_{t,i}(\sigma'), x_{t+1,i}(\sigma')\right) \geq \max(x_{t,i}(\sigma), x_{t+1,i}(\sigma)) - \frac{1}{10}|x_{t,i}(\sigma) - x_{t+1,i}(\sigma')| - \frac{3\alpha}{100Ld}$$

*Proof.* We have that

$$\sum_{i=1}^{t-1} f_i(x_t(\sigma)) - \langle \sigma, x_t(\sigma) \rangle + f_t(x_t(\sigma)) + m_{t+1}(x_t(\sigma))$$

$$\leq \sum_{i=1}^{t-1} f_i(x_{t+1}(\sigma)) - \langle \sigma, x_{t+1}(\sigma) \rangle + f_t(x_{t+1}(\sigma)) + +m_{t+1}(x_t(\sigma)) + \alpha$$

$$\leq \sum_{i=1}^{t-1} f_i(x_{t+1}(\sigma)) - \langle \sigma, x_{t+1}(\sigma) \rangle + f_t(x_{t+1}(\sigma)) + m_{t+1}(x_{t+1}(\sigma))$$

$$+ L\|x_{t+1}(\sigma) - x_t(\sigma)\|_1 + \alpha$$

Here, we have the first inequality from the approximate optimality of $x_t(\sigma)$ and the second inequality from the smoothness of the optimistic function. Moreover, from the opposite direction, we have that

$$\sum_{i=1}^{t-1} f_i(x_t(\sigma)) - \langle \sigma, x_t(\sigma) \rangle + f_t(x_t(\sigma)) + m_{t+1}(x_t(\sigma))$$

$$= \sum_{i=1}^{t-1} f_i(x_t(\sigma)) - \langle \sigma', x_t(\sigma) \rangle + \langle \sigma' - \sigma, x_t(\sigma) \rangle + f_t(x_t(\sigma)) + m_{t+1}(x_t(\sigma))$$

$$\geq \sum_{i=1}^{t-1} f_i(x_t(\sigma')) - \langle \sigma', x_{t+1}(\sigma') \rangle + \langle \sigma' - \sigma, x_t(\sigma) \rangle + f_t(x_{t+1}(\sigma'))$$

$$+ m_{t+1}(x_{t+1}(\sigma')) - \alpha$$

$$= \sum_{i=1}^{t-1} f_i(x_t(\sigma')) - \langle \sigma, x_{t+1}(\sigma') \rangle + \langle \sigma' - \sigma, x_t(\sigma) - x_{t+1}(\sigma') \rangle + f_t(x_{t+1}(\sigma'))$$

$$+ m_{t+1}(x_{t+1}(\sigma')) - \alpha$$

$$\geq \sum_{i=1}^{t-1} f_i(x_t(\sigma)) - \langle \sigma, x_{t+1}(\sigma) \rangle + \langle \sigma' - \sigma, x_t(\sigma) - x_{t+1}(\sigma') \rangle + f_t(x_{t+1}(\sigma))$$

$$+ m_{t+1}(x_{t+1}(\sigma)) - 2\alpha$$

Here, we have the first inequality from the approximate optimality of $x_t(\sigma')$ and the final inequality from the approximate optimality of $x_t(\sigma)$. From these and our original assumption, we have that,

$$10Ld\|x_{t+1,i}(\sigma) - x_{t,i}(\sigma)\|_1 + \alpha \geq 100Ld(x_{t,i}(\sigma) - x_{t+1,i}(\sigma')) - 2\alpha.$$

Using a similar argument, we have that

$$10Ld\|x_{t+1,i}(\sigma) - x_{t,i}(\sigma)\|_1 + \alpha \geq 100Ld(x_{t+1,i}(\sigma) - x_{t,i}(\sigma')) - 2\alpha.$$

Moreover, from Lemma E.2, we know that $x_{t+1,i}(\sigma') - x_{t+1,i}(\sigma) \geq \frac{-3\alpha}{100Ld}$ and $x_{t,i}(\sigma') - x_{t,i}(\sigma) \geq \frac{-3\alpha}{100Ld}$. Combining these, we have our claim. $\qquad\square$

We can now finally prove our claim. This proof is very similar to the proof of Theorem 1 in Suggala & Netrapalli (2019).

*Proof.* We note that we can decompose the $\ell_1$ norm in $\mathbb{E}\left[\|\mathbf{x}_t(\sigma) - \mathbf{x}_{t+1}(\sigma)\|_1\right]$ as

$$\mathbb{E}\left[\|\mathbf{x}_t(\sigma) - \mathbf{x}_{t+1}(\sigma)\|_1\right] = \sum_{i=1}^{d} \mathbb{E}\left[|\mathbf{x}_{t,i}(\sigma) - \mathbf{x}_{t+1,i}(\sigma)|\right].$$

To bound $\mathbb{E}\left[\|\mathbf{x}_t(\sigma) - \mathbf{x}_{t+1}(\sigma)\|_1\right]$ we derive an upper bound for each dimension $\mathbb{E}\left[|\mathbf{x}_{t,i}(\sigma) - \mathbf{x}_{t+1,i}(\sigma)|\right], \forall i \in [d]$. For any $i \in [d]$, define $\mathbb{E}_{-i}\left[|\mathbf{x}_{t,i}(\sigma) - \mathbf{x}_{t+1,i}(\sigma)|\right]$ as

$$\mathbb{E}_{-i}\left[|\mathbf{x}_{t,i}(\sigma) - \mathbf{x}_{t+1,i}(\sigma)|\right] = \mathbb{E}\left[|\mathbf{x}_{t,i}(\sigma) - \mathbf{x}_{t+1,i}(\sigma)| \mid \{\sigma_j\}_{j \neq i}\right]$$

where $\sigma_j$ is the $j^{\text{th}}$ coordinate of $\sigma$. Intuitively, we are computing in expectation of the noise of a single dimension while holding the other dimensions' noise constant. Let $\mathbf{x}_{\max,i}(\sigma) = \max(\mathbf{x}_{t,i}(\sigma), \mathbf{x}_{t+1,i}(\sigma))$ and $\mathbf{x}_{\min,i}(\sigma) = \min(\mathbf{x}_{t,i}(\sigma), \mathbf{x}_{t+1,i}(\sigma))$. Then, by definition, we have that

$$\mathbb{E}_{-i}\left[|\mathbf{x}_{t,i}(\sigma) - \mathbf{x}_{t+1,i}(\sigma)|\right] = \mathbb{E}_{-i}\left[\mathbf{x}_{\max,i}(\sigma)\right] - \mathbb{E}_{-i}\left[\mathbf{x}_{\min,i}(\sigma)\right].$$

Define event $\mathcal{E}$ as

$$\mathcal{E} = \{\sigma : \|\mathbf{x}_t(\sigma) - \mathbf{x}_{t+1}(\sigma)\|_1 \leq 10d \cdot |\mathbf{x}_{t,i}(\sigma) - \mathbf{x}_{t+1,i}(\sigma)|\}$$

For notational ease, let $\mathbf{P} = \exp(-100\eta Ld)$ be a constant. Consider the following

$$\begin{aligned}
\mathbb{E}_{-i}\left[\mathbf{x}_{\min,i}(\sigma)\right] = {} & \mathbb{P}\left(\sigma_i < 100Ld\right)\mathbb{E}_{-i}\left[\mathbf{x}_{\min,i}(\sigma) \mid \sigma_i < 100Ld\right] \\
& + \mathbb{P}\left(\sigma_i \geq 100Ld\right)\mathbb{E}_{-i}\left[\mathbf{x}_{\min,i}(\sigma) \mid \sigma_i \geq 100Ld\right] \\
\geq {} & (1 - \mathbf{P})\left(\mathbb{E}_{-i}\left[\mathbf{x}_{\max,i}(\sigma)\right] - D\right) \\
& + \mathbf{P}\mathbb{E}_{-i}\left[\mathbf{x}_{\min,i}\left(\sigma + 100Ld\mathbf{e}_i\right)\right]
\end{aligned}$$

where the last inequality follows the fact that the domain of $i^{\text{th}}$ coordinate lies within some interval of length $D$ and since $\mathbb{E}_{-i}\left[\mathbf{x}_{\min,i}(\sigma) \mid \sigma_i < 100Ld\right]$ and $\mathbb{E}_{-i}\left[\mathbf{x}_{\max,i}(\sigma)\right]$ are points in this interval, their difference is bounded by $D$. We can further lower bound $\mathbb{E}_{-i}\left[\mathbf{x}_{\min,i}(\sigma)\right]$ as follows

$$\begin{aligned}
\mathbb{E}_{-i}\left[\mathbf{x}_{\min,i}(\sigma)\right] \geq {} & (1 - \mathbf{P})\left(\mathbb{E}_{-i}\left[\mathbf{x}_{\max,i}(\sigma)\right] - D\right) \\
& + \mathbf{P}\mathbb{P}_{-i}(\mathcal{E})\mathbb{E}_{-i}\left[\mathbf{x}_{\min,i}\left(\sigma + 100Ld\mathbf{e}_i\right) \mid \mathcal{E}\right] \\
& + \mathbf{P}\mathbb{P}_{-i}\left(\mathcal{E}^c\right)\mathbb{E}_{-i}\left[\mathbf{x}_{\min,i}\left(\sigma + 100Ld\mathbf{e}_i\right) \mid \mathcal{E}^c\right]
\end{aligned}$$

where $\mathbb{P}_{-i}(\mathcal{E})$ is defined as $\mathbb{P}_{-i}(\mathcal{E}) := \mathbb{P}\left(\mathcal{E} \mid \{\sigma_j\}_{j \neq i}\right)$. We now use the monotonicity properties proved in Lemma E.2 and Lemma E.3 to further lower bound $\mathbb{E}_{-i}\left[\mathbf{x}_{\min,i}(\sigma)\right]$. Then

$$\begin{aligned}
\mathbb{E}_{-i}\left[\mathbf{x}_{min,i}(\sigma)\right] \geq {} & (1 - \mathbf{P})\left(\mathbb{E}_{-i}\left[\mathbf{x}_{\max,i}(\sigma)\right] - D\right) \\
& + \mathbf{P}\mathbb{P}_{-i}(\mathcal{E})\mathbb{E}_{-i}\left[\mathbf{x}_{\max,i}(\sigma) - \frac{1}{10}|\mathbf{x}_{t,i}(\sigma) - \mathbf{x}_{t+1,i}(\sigma)| - \frac{3\alpha}{100Ld} \mid \mathcal{E}\right] \\
& + \mathbf{P}\mathbb{P}_{-i}\left(\mathcal{E}^c\right)\mathbb{E}_{-i}\left[\mathbf{x}_{\min,i}(\sigma) - \frac{2\alpha}{100Ld} \mid \mathcal{E}^c\right] \\
\geq {} & (1 - \mathbf{P})\left(\mathbb{E}_{-i}\left[\mathbf{x}_{\max,i}(\sigma)\right] - D\right) \\
& + \mathbf{P}\mathbb{P}_{-i}(\mathcal{E})\mathbb{E}_{-i}\left[\mathbf{x}_{\max,i}(\sigma) - \frac{1}{10}|\mathbf{x}_{t,i}(\sigma) - \mathbf{x}_{t+1,i}(\sigma)| - \frac{3\alpha}{100Ld} \mid \mathcal{E}\right] \\
& + \mathbf{P}\mathbb{P}_{-i}\left(\mathcal{E}^c\right)\mathbb{E}_{-i}\left[\mathbf{x}_{\max,i}(\sigma) - \frac{1}{10d}\|\mathbf{x}_t(\sigma) - \mathbf{x}_{t+1}(\sigma)\|_1 - \frac{2\alpha}{100Ld} \mid \mathcal{E}^c\right]
\end{aligned}$$

where the first inequality follows from Lemma E.2 and Lemma E.3, the second inequality follows from the definition of $\mathcal{E}^c$. Rearranging the terms in the RHS and using $\mathbb{P}_{-i}(\mathcal{E}) \leq 1$ gives us

$$\begin{aligned}
\mathbb{E}_{-i}\left[\mathbf{x}_{\min,i}(\sigma)\right] \geq {} & (1 - \mathbf{P})\left(\mathbb{E}_{-i}\left[\mathbf{x}_{\max,i}(\sigma)\right] - D\right) \\
& + \mathbf{P}\mathbb{E}_{-i}\left[\mathbf{x}_{\max,i}(\sigma) - \frac{3\alpha}{100Ld}\right] \\
& - \mathbf{P}\mathbb{E}_{-i}\left[\frac{1}{10}|\mathbf{x}_{t,i}(\sigma) - \mathbf{x}_{t+1,i}(\sigma)| + \frac{1}{10d}\|\mathbf{x}_t(\sigma) - \mathbf{x}_{t+1}(\sigma)\|_1\right] \\
\geq {} & \mathbb{E}_{-i}\left[\mathbf{x}_{\max,i}(\sigma)\right] - 100\eta LdD - \frac{3\alpha}{100Ld} \\
& - \mathbb{E}_{-i}\left[\frac{1}{10}|\mathbf{x}_{t,i}(\sigma) - \mathbf{x}_{t+1,i}(\sigma)| + \frac{1}{10d}\|\mathbf{x}_t(\sigma) - \mathbf{x}_{t+1}(\sigma)\|_1\right]
\end{aligned}$$

where the last inequality uses the the fact that $\exp(x) \geq 1 + x$. Rearranging the terms in the last inequality gives us

$$\mathbb{E}_{-i}\left[|\mathbf{x}_{t,i}(\sigma) - \mathbf{x}_{t+1,i}(\sigma)|\right] \leq \frac{1}{9d}\mathbb{E}_{-i}\left[\|\mathbf{x}_t(\sigma) - \mathbf{x}_{t+1}(\sigma)\|_1\right] + \frac{1000}{9}\eta LdD + \frac{\mathbb{E}_{-i}[\alpha]}{30Ld}.$$

Since the above bound holds for any $\{\sigma_j\}_{j \neq i}$, we get the following bound on the unconditioned expectation

$$\mathbb{E}\left[|\mathbf{x}_{t,i}(\sigma) - \mathbf{x}_{t+1,i}(\sigma)|\right] \leq \frac{1}{9d}\mathbb{E}\left[\|\mathbf{x}_t(\sigma) - \mathbf{x}_{t+1}(\sigma)\|_1\right] + \frac{1000}{9}\eta LdD + \frac{\mathbb{E}[\alpha]}{30Ld}$$

Substituting this with the above yields the following bound on the stability of predictions of FTPL+

$$\mathbb{E}\left[\|\mathbf{x}_t(\sigma) - \mathbf{x}_{t+1}(\sigma)\|_1\right] \leq 125\eta Ld^2 D + \frac{\alpha}{20L}$$

Plugging the above bound gives us the required bound on regret. $\qquad\square$

## E.2 Proofs for Strong Gradient Dominance

We will first prove what the gradient of the value function is with respect to $\theta$ in this setting.

**Lemma 7.2.** *We have $g(W,\pi) = V_W^\pi(\mu) - \|\pi\|_2^2$ is $\mathcal{K}_\pi, \frac{T}{2}$-strongly-Gradient Dominated on set $\mathcal{T}$*

*Proof.* By the Performance Difference Lemma,

$$\sum_t V_{W_t}^{\pi^*} - \sum_t V_{W_t}^\pi - \langle \sigma, \pi^* - \pi \rangle - \|\pi^*\|_2^2 + \|\pi\|_2^2$$

$$= \frac{1}{1-\gamma}\sum_t\sum_{s,a} d_\mu^{\pi^*}(s)\pi^*(a,s)A^\pi(s,a) - \langle \sigma, \pi^* - \pi \rangle - \|\pi^*\|_2^2 + \|\pi\|_2^2$$

$$\leq \max_{\bar{\pi}}\left[\frac{1}{1-\gamma}\sum_t\sum_{s,a} d_\mu^{\pi^*}(s)\bar{\pi}(s,a)A^\pi(s,a) - \langle \sigma, \pi^* - \pi \rangle - \|\pi^*\|_2^2 + \|\pi\|_2^2\right]$$

$$\leq \left\|\frac{d_\mu^{\pi^*}}{d_\mu^\pi}\right\|_\infty \max_{\bar{\pi}}\left[\frac{1}{1-\gamma}\sum_t\sum_{s,a} d_\mu^\pi(s)\bar{\pi}(s,a)A^\pi(s,a) - \langle \sigma, \bar{\pi} - \pi \rangle - \|\bar{\pi}\|_2^2 + \|\pi\|_2^2\right]$$

$$= \left\|\frac{d_\mu^{\pi^*}}{d_\mu^\pi}\right\|_\infty \max_{\bar{\pi}}\left[\frac{1}{1-\gamma}\sum_t\sum_{s,a} d_\mu^\pi(s)(\bar{\pi}(s,a) - \pi(a,s))A^\pi(s,a) - \langle \sigma, \bar{\pi} - \pi \rangle\right.$$
$$\left. - \|\bar{\pi}\|_2^2 + \|\pi\|_2^2\right]$$

$$= \left\|\frac{d_\mu^{\pi^*}}{d_\mu^\pi}\right\|_\infty \max_{\bar{\pi}}\left[\frac{1}{1-\gamma}\sum_t\sum_{s,a} d_\mu^\pi(s)(\bar{\pi}(s,a) - \pi(a,s))Q^\pi(s,a) - \langle \sigma, \bar{\pi} - \pi \rangle\right.$$
$$\left. - \|\bar{\pi}\|_2^2 + \|\pi\|_2^2\right]$$

$$= \left\|\frac{d_\mu^{\pi^*}}{d_\mu^\pi}\right\|_\infty \max_{\bar{\pi}}\left[\frac{1}{1-\gamma}\sum_t\sum_{s,a} d_\mu^\pi(s)(\bar{\pi}(s,a) - \pi(a,s))Q^\pi(s,a) - \langle \sigma, \bar{\pi} - \pi \rangle\right.$$
$$\left. - \langle 2\pi, \bar{\pi} - \pi \rangle - \|\bar{\pi} - \pi\|_2^2\right]$$

$$\leq \frac{1}{1-\gamma}\left\|\frac{d_\mu^{\pi^*}}{\mu}\right\|_\infty \max_{\bar{\pi}}\left[(\bar{\pi} - \pi)^\top \nabla_\pi\left(\sum_t V_{W_t}^\pi - \langle \sigma, \pi \rangle - T\|\pi\|_2^2\right) - \frac{T}{2}\|\bar{\pi} - \pi\|_2^2\right]$$

Here, the first equality comes from the Performance Difference Lemma, the second equality comes from the fact that $\sum_a \pi(a,s)A^\pi(s,a) = 0$, the third equality comes from the definition of the

Advantage Function for the $\pi$-player, and the final inequality comes from the both the Policy Gradient Theorem and the fact that $d_\mu^\pi(s) \geq (1 - \gamma)\mu(s)$. Moreover, the second-from-last equality comes from the following logic

$$
\begin{aligned}
-\|\bar{\pi} - \pi\|_2^2 &= - \left[ \bar{\pi}^\top \bar{\pi} - 2\bar{\pi}^\top \pi + \pi^\top \pi \right] \\
&= - \left[ \bar{\pi}^\top \bar{\pi} - 2\bar{\pi}^\top \pi + 2\pi^\top \pi - \pi^\top \pi \right] \\
&= - \left[ \|\bar{\pi}\|_2^2 - \|\pi\|_2^2 + 2\langle \pi - \bar{\pi}, \pi \rangle \right] \\
&= -\|\bar{\pi}\|_2^2 + \|\pi\|_2^2 + 2\langle \bar{\pi} - \pi, \pi \rangle
\end{aligned}
$$

From this, we have that the value function done in this manner is strongly Gradient Dominated with constant $\frac{T}{2}$. $\qquad\square$

### E.3 PROOF OF THEOREM 6.1

**Theorem 6.1.** *Assume that the set of possible transition matrices $\mathcal{W}$ is convex. Let $L_\pi$ be the smoothness constant of $h_t$ with respect to the $\ell_1$ norm, $D_\pi = \max\limits_{\pi, \pi' \in \mathcal{T}} \|\pi - \pi'\|_2$, and $d_\pi$ be the dimension of the input for the $\pi$-player. Let $OL^\pi$ use FTPL and $OL^W$ use Best-Response. When $\eta = \frac{1}{L_\pi \sqrt{T d_\pi}}$ the robustness of the set of trained algorithms is for any $\bar{\pi}$*

$$
\min_{W^* \in \mathcal{W}} \sum_{t=0}^{T} V_{W^*}^{\bar{\pi}}(\mu) - \mathbb{E}\left[ \min_{W^* \in \mathcal{W}} \sum_{t=0}^{T} V_{W^*}^{\pi_t}(\mu) \right] \leq \frac{2 d_\pi^{\frac{3}{2}} D_\pi L_\pi}{\sqrt{T}} + c_\pi \left( T_{\mathcal{O}}, \mathcal{K}_\pi \right) + c_W \left( T_{\mathcal{O}}, \mathcal{K}_W \right).
$$

*Proof.* From Theorem 3.1, we have that

$$
\min_{W^*} \sum_{t=0}^{T} V_{W^*}^{\bar{\pi}}(\mu) - \min_{W^*} \sum_{t=0}^{T} V_{W^*}^{\pi_t}(\mu) \leq \operatorname{Reg}_W + \operatorname{Reg}_\pi.
$$

When the $\pi$-player is using FTPL, its regret is bounded according to

$$
\mathbb{E}(\operatorname{Reg}_\pi) \leq \mathcal{O}\left( \eta d_\pi^2 D_\pi L_\pi^2 + \frac{d_\pi D_\pi}{\eta T} + \alpha \right).
$$

Setting $\eta = \frac{1}{L_\pi \sqrt{T d_\pi}}$ to minimize this, we have

$$
\mathbb{E}(\operatorname{Reg}_\pi) \leq \mathcal{O}\left( \frac{2 d_\pi^{\frac{3}{2}} D_\pi L_\pi}{\sqrt{T}} + \alpha \right).
$$

Moreover, $\alpha$ is the Oracle Error term. Therefore, given that we have Gradient Dominance, using Projected Gradient Descent yields from Lemma 5.1

$$
\mathbb{E}(\operatorname{Reg}_\pi) \leq \mathcal{O}\left( \frac{2 d_\pi^{\frac{3}{2}} D_\pi L_\pi}{\sqrt{T}} + c_\pi \left( T_{\mathcal{O}}, \mathcal{K}_\pi \right) \right).
$$

Given the $W$-player employs Best-Response, we have that the regret of the $W$-player is bounded by the following by Lemma 4.1

$$
\mathbb{E}(\operatorname{Reg}_W) \leq \alpha
$$

where $\alpha$ is the optimization error. Given that we have Gradient Dominance properties for the $W$-player as well, we have that using the Projected Gradient Descent for

$$
\mathbb{E}(\operatorname{Reg}_W) \leq c_W \left( T_{\mathcal{O}}, \mathcal{K}_W \right).
$$

Adding these two inequalities together gets our final result. $\qquad\square$

### E.4 PROOF OF THEOREM 7.1

**Theorem 7.1.** *Assume that the set of possible transition matrices $\mathcal{W}$ is convex. Let $OL^\pi$ use OFTPL and $OL^W$ use FTPL+. Given that Condition 7.1 holds, we have that the robustness of the algorithm is for any $\bar{\pi}$ when setting $\eta = \sqrt{\frac{20L_\pi(d_w D_W + d_\pi D_\pi)^2}{d_\pi}D_\pi \tilde{L}_2 \alpha T}$,*

$$\min_{W^* \in \mathcal{W}} \sum_{t=0}^T V_{W^*}^{\bar{\pi}}(\mu) - \mathbb{E}\left[\min_{W^* \in \mathcal{W}} \sum_{t=0}^T V_{W^*}^{\pi_t}(\mu)\right] \le \mathcal{O}\left(\left[\frac{d_\pi \tilde{L}\sqrt{D_\pi(d_w D_w + d_\pi D_\pi)}}{\sqrt{5L_\pi T c_\pi(T_\mathcal{O}, K_\pi)}} + 1\right] c_\pi(T_\mathcal{O}, K_\pi) + c_W(T_\mathcal{O}, \mathcal{K}_W)\right).$$

*Proof.* From Theorem 3.1, we have that

$$\min_{W^*} \sum_{t=0}^T V_{W^*}^{\bar{\pi}}(\mu) - \min_{W^*} \sum_{t=0}^T V_{W^*}^{\pi_t}(\mu) \le \text{Reg}_W + \text{Reg}_\pi.$$

Given the $\pi$-player employs OFTPL, we have from Lemma D.1 that the regret is upper bounded by

$$\text{Reg}_\pi \le O\left(\eta d_\pi^2 D_\pi \sum_{t=1}^T \frac{L_t^2}{T} + \frac{d_\pi D_\pi}{\eta T} + \alpha\right).$$

However, we have that

$$\mathbb{E}(L_t^2) = \tilde{L}^2 \|W_t - W_{t-1}\|_1$$
$$\le \tilde{L}^2\left(125\eta L_\pi d_\pi^2 D_\pi + \frac{\alpha}{20L_\pi}\right)$$

Here, the last inequality comes from the fact that the $W$-player uses FTPL+, and the decisions made by FTPL+ are stable from Lemma E.1. Using this above, we have that

$$\text{Reg}_\pi \le O\left(\eta d_\pi^2 D_\pi \tilde{L}^2\left(125\eta L_\pi d_\pi^2 D_\pi + \frac{\alpha}{20L_\pi}\right) + \frac{d_\pi D_\pi}{\eta T} + \alpha\right).$$

Since our $\pi$-player enjoys Gradient Dominance for its loss function, we have from Lemma 5.1 that

$$\alpha \le c_\pi(T_\mathcal{O}, \mathcal{K}_\pi).$$

Moreover, given the $W$-player is employing FTPL+, from Lemma 4.2, we have that regret of the $W$-player is bounded by

$$\text{Reg}_W \le \mathcal{O}\left(\frac{d_W D_W}{\eta T} + \alpha\right).$$

In this setting, the $W$-player still enjoys Gradient Dominance properties, so using Projected Gradient Descent has

$$\alpha \le c_W(T_\mathcal{O}, \mathcal{K}_W).$$

Adding these together yields

$$\text{Reg}_\pi + \text{Reg}_W \le \mathcal{O}(\eta d_\pi^2 D_\pi \tilde{L}^2\left(125\eta L_\pi d_\pi^2 D_\pi + \frac{\alpha}{20L_\pi}\right) + \frac{d_W D_W + d_\pi D_\pi}{\eta T} +$$
$$c_\pi(T_\mathcal{O}, \mathcal{K}_\pi) + c_W(T_\mathcal{O}, \mathcal{K}_W)).$$

Setting $\eta = \sqrt{\frac{20L_\pi(d_w D_W + d_\pi D_\pi)^2}{d_\pi}D_\pi \tilde{L}_2 \alpha T}$, we have that

$$\text{Reg}_\pi + \text{Reg}_W \le \mathcal{O}\left(\left[\frac{d_\pi \tilde{L}\sqrt{D_\pi(d_w D_w + d_\pi D_\pi)}}{\sqrt{5L_\pi T c_\pi(T_\mathcal{O}, K_\pi)}} + 1\right] c_\pi(T_\mathcal{O}, K_\pi) + c_W(T_\mathcal{O}, \mathcal{K}_W)\right).$$

$\square$

### E.5 PROOF FOR THEOREM 7.2

**Theorem 7.2.** *Assume that the set of possible transition matrices $\mathcal{W}$ is convex. Let $OL^\pi$ use FTPL and $OL^W$ use Best-Response. Given that Condition 7.2 holds and $\eta = \frac{1}{L_\pi \sqrt{T d_\pi}}$, we have that the robustness of Algorithm 1 is for any $\bar{\pi}$*

$$\min_{W^* \in \mathcal{W}} \sum_{t=0}^{T} V_{W^*}^{\bar{\pi}}(\mu) - \|\bar{\pi}\|^2 - \min_{W^* \in \mathcal{W}} \sum_{t=0}^{T} V_{W^*}^{\pi_t}(\mu) + \|\pi_t\|^2 \leq \frac{2d_\pi^{\frac{3}{2}} D_\pi L_\pi}{\sqrt{T}} + c_\pi^s \left(T_\mathcal{O}, \mathcal{K}_\pi, \frac{1}{2}\right) + c_W(T_\mathcal{O}, \mathcal{K}_W).$$

*Proof.* From Theorem 3.1, we have that

$$\min_{W^*} \sum_{t=0}^{T} V_{W^*}^{\bar{\pi}}(\mu) - \min_{W^*} \sum_{t=0}^{T} V_{W^*}^{\pi_t}(\mu) \leq \text{Reg}_W + \text{Reg}_\pi.$$

When the $\pi$-player is using FTPL, its regret is bounded according to

$$\mathbb{E}(\text{Reg}_\pi) \leq \mathcal{O}\left(\eta d_\pi^2 D_\pi L_\pi^2 + \frac{d_\pi D_\pi}{\eta T} + \alpha\right).$$

Setting $\eta = \frac{1}{L\sqrt{T d}}$ to minimize this, we have

$$\mathbb{E}(\text{Reg}_\pi) \leq \mathcal{O}\left(\frac{2d_\pi^{\frac{3}{2}} D_\pi L_\pi}{\sqrt{T}} + \alpha\right).$$

Moreover, $\alpha$ is the Oracle Error term. Therefore, given that we have strong Gradient Dominance, using Projected Gradient Descent yields from Lemma 7.1

$$\mathbb{E}(\text{Reg}_\pi) \leq \mathcal{O}\left(\frac{2d_\pi^{\frac{3}{2}} D_\pi L_\pi}{\sqrt{T}} + + c_\pi^s \left(T_\mathcal{O}, \mathcal{K}_\pi, \frac{1}{2}\right)\right).$$

Given the $W$-player employs Best-Response, we have that the regret of the $W$-player is bounded by the following by Lemma 4.1

$$\mathbb{E}(\text{Reg}_W) \leq \alpha$$

where $\alpha$ is the optimization error. Given that we have Gradient Dominance properties for the $W$-player as well, we have that using the Projected Gradient Descent for

$$\mathbb{E}(\text{Reg}_W) \leq c_W(T_\mathcal{O}, \mathcal{K}_W).$$

Adding these two inequalities together gets our final result. □

