# OpenReview forum: "Solving Robust MDPs through No-Regret Dynamics"
_ICLR.cc/2024/Conference — ICLR 2024 Conference Withdrawn Submission_

### Official Review · Reviewer_ZrbN · 2023-10-30

**Soundness:** 2 fair
**Presentation:** 2 fair
**Contribution:** 2 fair
**Rating:** 3
**Confidence:** 3

**Summary:**

This paper propose a non-convex no-regret framework for robust MDPs by solving a minimax iterative optimization problem where two players (agent and environment) play against each other. With a convex uncertainty set assumption, which relaxes the commonly used rectangular assumption, they devise several algorithms solve robust MDPs on the order of $\mathcal{O}(1/T^{1/2})$.

**Strengths:**

1. A novel framework solving robust MDPs .

2. Two new non-convex online learning algorithms designed specifically for the environmental player.

3. A $1/\sqrt{T}$ convergence rate for their algorithm. Relaxation of specific rectangularity uncertainty set assumptions to a general convex uncertainty set assumption.

**Weaknesses:**

1. Some places need clarification.
2. This paper aims to provide a general and efficient theoretical Analysis for robust MDPs, however, it needs strong assumption of Gradient Dominance, which may not hold in many cases. This weakens the theoretical contribution of the framework.
3. Computation may not be efficient.

**Questions:**

1. What are the two average regrets $Reg_W$ and $Reg_\pi$? No definition in the main context.

2. Why a noise vector is incorporated in solving $x_t=argmin_{x_t}\sum_{i=1}^tf_i(x_t)-\langle\sigma,x_t \rangle$?

3. At the bottom of page 5, what is the OFTPL and what is the optimistic function $m_t$?

4. Definition 5.1is problematic, as the LHS $\geq 0$ while the RHS $\leq 0$.

5. Could you please further explain why we need those conditions in Condition 5.1?

6. The experiment is too simple to illustrate the computational performance. More discussions about computation are needed. How to implement those algorithms efficiently, especially with general convex uncertainty sets. How about the computation complexity? How it performs compared with state-of-the-art algorithm designed for rectangularity uncertainty set (as they are all convex)?

7. As your goal is to develop a general and efficient theoretical analysis for robust MDP, it seems that the only advantage compared to existing analysis is the relaxation of rectangular assumptions. However, your theory introduce addition strong assumptions. Could you highlight the advantage of your framework? Does the current framework and theoretical results cover existing works?

---

### Official Review · Reviewer_Qfzd · 2023-11-04

**Soundness:** 3 good
**Presentation:** 2 fair
**Contribution:** 2 fair
**Rating:** 3
**Confidence:** 3

**Summary:**

This paper provides a novel online learning perspective for solving *known* (i.e., the planning problem of) robust MDPs. It proposes a no-regret framework that characterizes robust optimization as a min-max game and establishes a universal sub-optimality bound using regrets of both players. Then it introduces a few online optimization algorithms based on known *approximate optimization oracles $\mathcal{O}_{\alpha}$*, and applies *projected gradient descent* to construct such oracles given *gradient dominated* objective functions. Finally, for a specific example, it shows the required gradient dominance property for the value function $V_W^{\pi}(\mu)$ holds under *direct parametrization* of models, which immediately results in a sub-optimality bound according to the framework.

**Strengths:**

1. This paper takes an interesting angle to promote the understanding of robust MDPs. The reviewer believes this is among the first to apply FTL-ish algorithms to solve robust MDPs, probably inspired by existing similar works for adversarial MDPs.

2. After the reviewer's *careful verification*, the mathematical proofs seem to *contain the correct ideas* in the form they are presented in the paper, though there are quite a few typos and gaps that obstructs the reading.

**Weaknesses:**

1. The writing of this paper needs to be improved. The connections between sections are weak (a single sentence in most sections), and the techniques are not introduced in a motivated way. This makes the contribution of this paper most unclear to a first-time reader — what is the ultimate problem that this paper attempts to solve, and to what extent has it been solved? What can be done with it beyond standard value function and direct parametrization?

2. The authors need to be more careful about the math used in the paper, both in the statement of definitions/theorems and the proofs. Here are a few obvious typos/gaps that confuse the reviewer for quite some time:
    * The definition of gradient dominance is inconsistent with [Bhandari & Russo, 2022]. The RHS term should be negative. Otherwise, since $x \in \mathcal{X}$, we immediately have $f(x) - \min_{x^* \in \mathcal{X}} f(x^*) \leq 0$, which is ridiculous. Fortunately the subsequent lemmas carry the correct sign.
    * Lemma 5.2 is stated in the wrong way. There should not be a $\mathbb{P}_W$ factor in the summand.
    * It needs to be emphasized that Lemma 5.4 and 5.5 only hold for direct parametrization, which is not reflected in the proof (e.g., in the third set of equations on page 15, the equation $\mathbb{P}\_W(s,a,s') = W\_{s,a,s'}$ should at least show up).
    * There is an additional $\gamma^t$ factor in the third set of equations on page 15, making it slightly different from what is established in Lemma 5.2.

The reviewer urges the authors to double check all statements and proofs. It is also advisable to number the equations for the convenience of reference.

3. Though it contains a series of analyses and proofs, the contribution of this paper is suspicious.
    * There is not sufficient literature review that helps to establish the contribution and novelty of this paper. Section 1 is way too short for a well-studied area.
    * This is basically a planning setting with no learning involved. The reviewer doubts whether the proposed framework can be extended to deal with unknown MDPs for the learning setting.
    * It seems to the reviewer that this is just a concatenation of known results — the formulation of the solution of robust MDPs as a min-max game is well-known, and the algorithms are standard online for optimization.
    * The reviewer thinks that this paper might be abstracting straight-forward results in an unnecessary way. It's hard to imagine a reward/loss function $g(W, \pi)$ other than $V_W^{\pi}$. Plus, although claimed by the authors, it is unclear how similar gradient dominance properties could be established for generic parametrization of the model.
    *  If the framework only works for direct parametrization (as it turns out to be in this paper), the underlying MDP can only be tabular, and scalability would be an issue (in which case $d_{\pi} = S^2 A$ is potentially large).

The reviewer is open to raising the rating if the typos and gaps are fixed, and the concerns regarding contribution are settled.

**Questions:**

See the weakness part.

---

### Official Review · Reviewer_fufQ · 2023-11-05

**Soundness:** 2 fair
**Presentation:** 3 good
**Contribution:** 1 poor
**Rating:** 3
**Confidence:** 3

**Summary:**

The authors consider the setting of robust MDPs. The problem is formalized as a minmax problem. In turn, the authors attempt to leverage online nonconvex no-regret algorithms in order to provide improved convergence bounds.
The paper is burdened by multiple typos which I would be able to disregard had I not found an error that in my opinion is unfixable and lies in the heart of the arguments used to prove the main theorem.

In particular, the proof of Lemma 5.4 not only is ridden with typos, it also has an important error. The advantage function can very well be negative, this means that the first inequality does not hold! Unless the authors can actually prove lemma 5.4, I don't think the paper can be considered for acceptance.

More comments:
The statement of lemma 5.3 seems to have a typo, but so does its proof. Proof of Lemma 5.3 seems to include a typo that is present in the arxiv version of "On the Theory of Policy Gradient Methods: Optimality, Approximation, and Distribution Shift".
It seems like the definition of Condition 7.2 has a typo. Should it not be $\| x' - x\|^2$? Authors should also cite the precise definitions and theorems they invoke from Karimi et al.

**Strengths:**

The strength of the paper is the clear presentation of the topic and the proof of additional properties holding with minimal assumptions (i.e., the convexity of the uncertainty set of matrices $\mathbb{P}$. Indeed, the authors could have leveraged a different algorithm than FPTL and get some improved results.

**Weaknesses:**

The weakness of the paper lies in the multiple typos and the error in the proof of lemma 5.4 I already highlighted.

**Questions:**

Do you think you could fix the proof of lemma 5.4?

---

### Official Review · Reviewer_yiTS · 2023-11-05

**Soundness:** 3 good
**Presentation:** 1 poor
**Contribution:** 2 fair
**Rating:** 3
**Confidence:** 4

**Summary:**

This paper leverages techniques from no regret learning, and generalize some techniques to non-convex setting so that they could be used to solve robust markov decision processes. The unique feature of the proposed approach is that it does not rely on assumption of rectangularity, which is commonly assumed in the robust markov decision process.

**Strengths:**

I think this paper aims to address an important problem by using an innovative approach that was not considered before. The issue of sensitivity is important and well-known in RL, and so solving robust RL appears to be an interesting alternative to existing SOTA.

**Weaknesses:**

My biggest concern of this paper is the presentation. In my honest opinion, this paper is poorly written with messy notation. I understand that the author(s) is trying to provide many information under different setups, but the current form of this paper is not friendly for readers who do not have a strong background in related topics. For example, when reading this paper, sometimes the author(s) refer to objective V, sometimes g, and sometimes f. I think there is a large room for improvement in terms of the writing.

Moreover, I think the author(s) did not provide sufficient background and material to motivate their results. For example, what is the problem of rectangularity and why the proposed algorithm can avoid this assumption. Why is that a critical assumption? and how it would affect the agent's performance. Currently this paper lacks of good experiments, and it would be great if the authors can answer these questions and highlight their contributions from both theoretical and experimental perspectives.

Also, it seems that the authors only provide an outline of the proposed algorithms, but they did not describe in details how robust markov decision process can actually be solved. It would be nice if there are more information about the entire algorithm and provide the time and space complexity.

Last but not least, I am not totally convinced with the proposed approach of using no regret learning. To the best of my understanding, the purpose of solving robust markov decision processes is to avoid taking risky actions. Also, it is not clear to me how the environment can follow the proposed algorithm in practice; as an agent, I don't think one can control the environment?

**Questions:**

What the space and time complexities for each step in Algorithm 1?

What are the mathematical definitions of Reg_W and Reg_pi?

Section 4, what is the loss function f_i, in the context of robust markov decision process?

Theorem 6.1, do c_pi and c_w converges to 0 as T goes to infinity?

---

### Official Review · Reviewer_MYdb · 2023-11-06

**Soundness:** 2 fair
**Presentation:** 2 fair
**Contribution:** 2 fair
**Rating:** 3
**Confidence:** 3

**Summary:**

In this paper, the authors provide a framework to solve the Robust MDP setting. They developed two robust online learning algorithms.

**Strengths:**

- The paper provides two algorithms to solve a relevant problem, i.e., the Robust MDP setting.

**Weaknesses:**

- Clarity: the authors can work to improve the clarity and the writing style of the paper. The paper is not formal enough and some parts are a bit confusing:

    - Preliminary:

        - It is not clear how $W$ is defined and which kind of parametrization is considered.

        - Where is $\gamma$? Which setting is considered? Finite-horizon, infinite-horizon, average reward etc.?

        - It would be better to define the value function in the notation. part since it has already been introduced there.

        - The state distribution frequency is the same $d^W_{s_0}$ and $d^\pi_{s_0}$ but it would be better to be function of both $W$ and $\pi$.

        - Lines after Definition 2.2. are not relevant since the repeated games vision is not used later in the paper.

   - No Regret Dynamics

        - What is the regret that are we trying to minimize? And why is it relevant to consider this definition for the Robust MDP setting?

        - What is the meaning of OL$^\pi$ (OL$^W$)?

        - Why do the two loss functions depend only on one variable each?

   -  Regret Minimization for Smooth Nonconvex Loss Functions

        - What is the final algorithm used in this section?

        - "regret bond" --> "regret bound"

        - Some phrases need more context (e.g. This will be useful for later extensions)

        - Lemma 4.1-4.2: It would be better to specify what the optimization oracle is computing

        - Lemma 4.2: Gradient Dominated loss is not still defined

        - It would be better to have an explanation of the algorithms from Suggala & Netrapalli.

   -  Gradient Dominance:

        - It could be better to choose a different name rather than item 4

    - Extensions:

        - Many equations are outside the border of the paper.

- Novelty:

     - It is not very clear what is the novelty of the approach. The no-Regret algorithms such as Follow the Regularized leader are well-known algorithms.

    - What are the technical challenges of the proposed approach?

- Comparison with SotA:

   - What are the differences between the proposed approach and the SotA?

   - What are the previous regret guarantees on the robust MDP setting considered?

- Experimental evaluation:

   - It would be good to add the experimental evaluation to the main paper to show what are the advantages of the proposed algorithms with respect to previous approaches.

**Questions:**

See weaknesses.